# Finding Low-Rank Matrix Weights in DNNs via Riemannian Optimization: RAdaGrad and RAdamW

**Fengmiao Bian**[1], **Jinyang Zheng**[1], **Ziyun Liu**[1], **Jianzhou Luo**[1], **Jian-Feng Cai**[1*]

[1] Department of Mathematics
The Hong Kong University of Science and Technology, Hong Kong, CHINA

`{mafmbian,jfcai}@ust.hk,{jzhengbp,zliueq,jluobn}@connect.ust.hk`

## Abstract

Finding low-rank matrix weights is a key technique for addressing the high memory usage and computational demands of large models. Most existing algorithms rely on the factorization of the low-rank matrix weights, which is non-unique and redundant. Their convergence is slow especially when the target low-rank matrices are ill-conditioned, because the convergence rate depends on the condition number of the Jacobian operator for the factorization and the Hessian of the loss function with respect to the weight matrix. To address this challenge, we adopt the Riemannian gradient descent (RGD) algorithm on the Riemannian manifold of fixed-rank matrices to update the entire low-rank weight matrix. This algorithm completely avoids the factorization, thereby eliminating the negative impact of the Jacobian condition number. Furthermore, by leveraging the geometric structure of the Riemannian manifold and selecting an appropriate metric, it mitigates the negative impact of the Hessian condition number. Ultimately, this results in our two plug-and-play optimizers: RAdaGrad and RAdamW, which are RGD with metrics adapted from AdaGrad and AdamW and restricted to the manifold. Our algorithms can be seamlessly integrated with various deep neural network architectures without any modifications. We evaluate the effectiveness of our algorithms through fine-tuning experiments on large language models and diffusion models. Experimental results consistently demonstrate that our algorithms provide superior performance compared to state-of-the-art methods. Additionally, our algorithm is not only effective for fine-tuning large models but is also applicable to deep neural network (DNN) compression.

## 1 Introduction

Deep Neural Networks (DNNs) have achieved remarkable success in tasks such as image classification [19], object detection [20], and semantic segmentation [8]. However, their high memory and computational demands pose significant challenges for deployment on resource-constrained devices. For example, ResNet-50 requires up to 4G FLOPs for a single image classification task [6], making it unsuitable for embedded systems. To address these limitations, techniques such as pruning [17, 48] and quantization [28, 44] have been developed. While effective, these methods often rely on specialized hardware for acceleration, limiting their general applicability.

Low-rank matrix approximation has emerged as a promising alternative, widely used in DNN compression [30, 37, 43], fine-tuning of large models (LMs) [18, 21, 47], and prompt engineering [15, 24]. By representing weight matrices as the product of two low-rank factors, this approach

---

*Correspondence to: Jian-Feng Cai<jfcai@ust.hk>.

reduces memory usage while preserving input and output dimensions, without requiring specialized hardware. In LMs, methods like Low-Rank Adaptation (LoRA) [18, 21] train additional low-rank matrix weights, significantly reducing computational costs while maintaining or even improving performance compared to full-parameter fine-tuning. These advantages make low-rank matrix methods indispensable for efficient adaptation of large models to downstream tasks.

Numerous methods have been proposed to obtain low-rank matrix weights in neural networks. Some approaches [11, 22, 27, 50] employ singular value decomposition (SVD) to directly compute low-rank matrices weight, but the frequent large-scale SVD operations are computationally expensive. More commonly, factorization-based methods [18, 21, 38, 47, 49, 51] factorize a rank-$r$ matrix weight $W \in \mathbb{R}^{m \times n}$ as $W = PQ^\top$, where $P \in \mathbb{R}^{m \times r}$ and $Q \in \mathbb{R}^{n \times r}$. While this approach reduces memory usage by storing $P$ and $Q$, it has several inherent drawbacks. First, the factorization $W = PQ^\top$ is not unique, introducing redundancy that can lead to unbalanced factors, slow convergence, or even failure to converge. Second, as we will demonstrate later, the convergence rate of these algorithms depends on the condition number of the Jacobian operator $\mathcal{J}_\mathcal{G}(P, Q)$ of the generator $\mathcal{G}(P, Q) = PQ^\top$. For ill-conditioned target matrices $W$, the condition number of $\mathcal{J}_\mathcal{G}(P, Q)$ becomes very large, significantly slowing convergence. Finally, efficient optimizers such as AdaGrad [12] and AdamW [26, 29], which are designed to optimize the full weight matrix $W$, cannot be directly applied to the low-rank factors $P$ and $Q$ in a way that fully exploits their efficiency.

To overcome these limitations, we propose a novel optimization framework based on Riemannian optimization. Specifically, we employ the Riemannian Gradient Descent (RGD) algorithm on the manifold of fixed-rank matrices to directly update the entire low-rank weight matrix $W$, rather than its low-rank factors $P$ and $Q$. While our method has the same order of computational complexity and memory usage as factorization-based methods, it offers significant advantages. First, it fully exploits the geometric structure of the Riemannian manifold and the intrinsic representation of low-rank matrices, avoiding the non-uniqueness and redundancy introduced by factorization. Second, by directly optimizing $W$, our method only involves $\nabla \ell(W)$ and is independent of the Jacobian operator $\mathcal{J}_\mathcal{G}(P, Q)$, making the convergence rate independent of the condition number of the target matrix $W$. Finally, the RGD algorithm allows us to define different metrics on the manifold, enabling the extension of classical adaptive learning rate and momentum algorithms, such as AdaGrad [12] and AdamW [26, 29], onto the Riemannian manifold. Building on this framework, we introduce two novel algorithms: RAdaGrad and RAdamW. These algorithms retain the adaptability of AdaGrad and AdamW while being better suited to the unique structure of low-rank matrix optimization problems.

Moreover, RAdaGrad and RAdamW function as plug-and-play optimizers that can seamlessly integrate into a wide range of deep neural network architectures without requiring any changes to the network structure. To evaluate the effectiveness of our approach, we conduct extensive experiments, including fine-tuning of large language models and diffusion models. The results consistently show that RAdaGrad and RAdamW significantly outperform state-of-the-art methods. Furthermore, our approach is not only highly effective for fine-tuning large models but also demonstrates great potential in deep neural network compression tasks, highlighting its versatility and practicality.

## 2 Finding Low-Rank Weights in DNNs

In this section, we examine existing algorithms, discuss their limitations, and propose a novel framework designed to overcome these challenges. To find the low-rank matrix weights of a deep neural network, we solve the following optimization problem

$$\min_W \ell(W), \ \text{subject to} \ \text{rank}(W^{(k)}) = r_k, \ k = 1, \ldots, K, \tag{1}$$

where $W = (W^{(1)}, W^{(2)}, \ldots, W^{(K)}) \in \mathbb{E} := \prod_{k=1}^K \mathbb{R}^{n_{k-1} \times n_k}$ represents the weight matrices with $W^{(k)} \in \mathbb{R}^{n_{k-1} \times n_k}$, $\ell(W)$ is the training loss function, and $r_k$ represents the rank of $W^{(k)}$. We use $\nabla \ell(W)$ to denote its gradient, or (batch) stochastic gradient, without ambiguity.

### 2.1 Factorization-Based Algorithms

Current mainstream low-rank adaptation methods [18, 21, 38, 47, 49, 51] for finding low-rank weights primarily rely on factorization-based algorithms, where each weight matrix $W^{(k)}$ is parameterized as $W^{(k)} = P^{(k)} (Q^{(k)})^\top$ with $P^{(k)} \in \mathbb{R}^{n_{k-1} \times r_k}$, $Q^{(k)} \in \mathbb{R}^{n_k \times r_k}$. Here, $\top$ represents matrix transpose.

Let $P = (P^{(1)}, P^{(2)}, \ldots P^{(K)})$ and $Q = (Q^{(1)}, Q^{(2)}, \ldots, Q^{(K)})$ be the factors, and define the generator

$$\mathcal{G}(P, Q) = (P^{(1)}(Q^{(1)})^\top, P^{(2)}(Q^{(2)})^\top, \ldots, P^{(K)}(Q^{(K)})^\top) \tag{2}$$

as the mapping from these factors to the weight matrices. Factorization-based algorithms obtain low-rank weights by solving

$$\min_{P,Q} f(P, Q), \quad \text{where} \quad f(P, Q) := \ell(\mathcal{G}(P, Q)). \tag{3}$$

The gradient of $f(P, Q)$ is given by the chain rule

$$\nabla f(P, Q) = \mathcal{J}_\mathcal{G}^*(P, Q) \cdot \nabla \ell(W) \text{ with } W = \mathcal{G}(P, Q), \tag{4}$$

where $\mathcal{J}_\mathcal{G}$ is the Jacobian operator of the operator $\mathcal{G}$ and $*$ denotes the adjoint of the operator. The Hessian is given by

$$\nabla^2 f(P, Q) = \mathcal{J}_\mathcal{G}^*(P, Q) \cdot \nabla^2 \ell(W) \cdot \mathcal{J}_\mathcal{G}(P, Q) + \text{the terms related to the changes in } \mathcal{J}_\mathcal{G}(P, Q). \tag{5}$$

The original LoRA algorithm [21] directly applies gradient descent to solve (1), and its convergence rate is influenced by the condition number of the Hessian $\nabla^2 f(P, Q)$ [4, 34]. From (5), it is evident that even when only the first term is considered, the convergence rate of the algorithm is significantly affected by the condition numbers of both the Jacobian operator $\mathcal{J}_\mathcal{G}(P, Q)$ and the Hessian $\nabla^2 \ell(W)$. Since the condition number of $\mathcal{J}_\mathcal{G}(P, Q)$ depends on the condition number of $W$ [13, 32], the convergence rate deteriorates significantly when the target matrix $W$ is ill-conditioned. Moreover, imbalances in the factorization of $W$ and poor initialization can also lead to slower convergence.

To develop efficient algorithms, it is therefore crucial to improve the condition of $\mathcal{J}_\mathcal{G}(P, Q)$ and $\nabla^2 \ell(W)$.

- Efforts to improve the condition of $\mathcal{J}_\mathcal{G}(P, Q)$: several recent attempts have been made to address this issue:
  - LoRA+[18] employs different learning rates for the two low-rank factors to improve convergence. However, it does not fully eliminate dependence on the condition number of $\mathcal{J}_\mathcal{G}(P, Q)$.
  - Riemannian preconditioned LoRA [47] applies simple preconditioners to each factor, partially mitigating but not entirely eliminating this dependence.
  - LoRA-RITE [46] LoRA-RITE employs preconditioners based on transformation invariance, alleviating the impact of the condition number of $\mathcal{J}_\mathcal{G}(P, Q)$.
  - LoRA-PRO [49] removes dependence on the condition number of $\mathcal{J}_\mathcal{G}(P, Q)$ by projecting factor-based gradients back to a subspace of the matrix space.
  - DLRT [38] eliminates this dependence by updating low-rank factors on the quotient manifold.
  - Imbalance-Regularized LoRA [51] reduces the impact through regularization terms applied to the low-rank factors.

- Addressing the condition of $\nabla^2 \ell(W)$: none of the aforementioned works directly address the condition number of $\nabla^2 \ell(W)$.
  - When the low-rank constraint is absent, there are well-established methods to address this issue. Classical optimizers such as AdaGrad and AdamW adaptively assign different step sizes to each component of $\nabla \ell(W)$. This essentially imposes an adaptive metric in the matrix space, under which the condition number of $\nabla^2 \ell(W)$ is smaller compared to the standard metric. However, extending this idea to low-rank factors presents significant challenges. Although it is possible to directly use AdaGrad or AdamW to solve (3) by assigning adaptive step sizes to each component of $\nabla f(P, Q)$, the redundancy in the parameterization of the low-rank factorization leads to poor algorithm performance, as demonstrated in our experimental section.

### 2.2 Riemannian Optimization Framework

To develop efficient algorithms, we propose using Riemannian optimization to solve (1), which addresses all the issues caused by the factorization-based methods discussed in Section 2.1.

It is well known that all rank-$r_k$ matrices form a smooth Riemannian manifold [5, 42], denoted as $\mathcal{M}_{r_k}$, embedded in the matrix space $\mathbb{R}^{n_{k-1} \times n_k}$. Consequently, the low-rank weight matrices $W$ belong to $\mathcal{M}_r := \prod_{k=1}^K \mathcal{M}_{r_k}$, and $\mathcal{M}_r$ is itself a smooth Riemannian manifold embedded in $\mathbb{E}$. Therefore, (1) can be reformulated as

$$\min_{W \in \mathcal{M}_r} \ell(W). \tag{6}$$

We solve this problem using the Riemannian Gradient Descent (RGD) algorithm

$$W_{t+1} = \mathcal{R}(W_t - \gamma_t \nabla_{\mathcal{M}_r} \ell(W_t)), \tag{7}$$

where $W_t \in \mathcal{M}_r$ represents the weight matrix at the $t$-th iteration, $\gamma_t \in \mathbb{R}_+$ denotes the step size, $\nabla_{\mathcal{M}_r} \ell(W_t) \in \mathbb{T}_{W_t}$ is the Riemannian gradient that lies in the tangent space $\mathbb{T}_{W_t}$ of the manifold $\mathcal{M}_r$ at $W_t$, and $\mathcal{R} : \mathbb{T}_{W_t} \to \mathcal{M}_r$ is a retraction operator that maps matrices from $\mathbb{T}_{W_t}$ back to $\mathcal{M}_r$. At iteration $t$, the low-rank weight matrices $W_t$ are updated on the tangent space using the Riemannian gradient, yielding

$$Z_t := W_t - \gamma_t \nabla_{\mathcal{M}_r} \ell(W_t) \in \mathbb{T}_{W_t}, \tag{8}$$

where $Z_t$ is then retracted back onto the manifold $\mathcal{M}_r$ using the retraction operator $\mathcal{R}$, resulting in the new low-rank weight matrices $W_{t+1}$. It is worth noting that different Riemannian metrics can be chosen to define the inner product on the tangent spaces of the Riemannian manifold. These metrics affect both the direction and magnitude of the Riemannian gradient, thereby giving rise to different algorithms. This provides a general framework where various optimization algorithms can be derived by selecting appropriate Riemannian metrics.

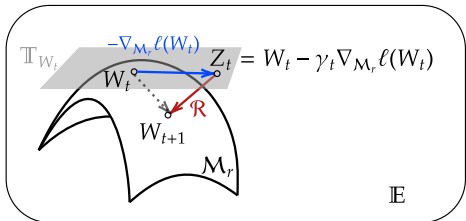

By calculation, the Riemannian gradient of $\ell(W)$ on $\mathcal{M}_r$ is given by

$$\nabla_{\mathcal{M}_r} \ell(W) = \mathcal{P}_{\mathbb{T}_W} \nabla \ell(W), \tag{9}$$

where $\mathcal{P}_{\mathbb{T}_W}$ is the orthogonal projection from $\mathbb{E}$ onto the tangent space $\mathbb{T}_W$, and $\nabla \ell(W)$ is the gradient of $\ell(W)$ with respect to $W$ in $\mathbb{E}$. The Riemannian Hessian of $\ell(W)$ on $\mathcal{M}_r$ is given by:

$$\nabla^2_{\mathcal{M}_r} \ell(W) = \mathcal{P}_{\mathbb{T}_W} \cdot \nabla^2 \ell(W) \cdot \mathcal{P}_{\mathbb{T}_W} + \text{terms related to the changes in } \mathcal{P}_{\mathbb{T}_W}, \tag{10}$$

where $\nabla^2 \ell(W)$ is the Hessian of $\ell(W)$ in $\mathbb{E}$. Here, all $\mathcal{P}_{\mathbb{T}_W}$, $\nabla \ell(W)$, and $\nabla^2 \ell(W)$ are computed under the metric on $\mathbb{T}_W$ extended to $\mathbb{E}$.

We will demonstrate in the next section that, with suitable choices of Riemannian metrics, the computational and memory complexities of RGD are the same order as those of factorization-based algorithms. Furthermore, RGD overcomes all the drawbacks of factorization-based algorithms mentioned in Section 2.1 and offers several advantages:

- *Avoiding Redundancy.* RGD operates on the manifold of fixed-rank matrices, directly updating the entire low-rank weight matrix $W$ rather than its factorized components $P$ and $Q$. By leveraging the geometric structure of the Riemannian manifold and the intrinsic representation of the tangent space, RGD avoids the non-uniqueness and redundancy introduced by factorization. Specifically, as demonstrated in [42, 45], it is easy to find an orthonormal basis for $\mathbb{T}_{W_t}$. This allows us to efficiently compute updates for $Z_t$ in (29) under the orthonormal basis of $\mathbb{T}_{W_t}$, maintaining the computational complexity and memory usage at the same order as factorization-based algorithms while eliminating redundancy.

- *Convergence Rate Depends Only on the Euclidean Hessian.* The convergence rate of RGD is primarily influenced by the condition number of $\nabla^2 \ell(W)$ only. It is well known that the convergence rate of RGD is affected by the condition number of $\nabla^2_{\mathcal{M}_r} \ell(W)$ [1]. From (10), if the second term is negligible, then $\nabla^2_{\mathcal{M}_r} \ell(W) \approx \mathcal{P}_{\mathbb{T}_W} \cdot \nabla^2 \ell(W) \cdot \mathcal{P}_{\mathbb{T}_W}$, which is the restriction of $\nabla^2 \ell(W)$ to $\mathbb{T}_W$. Hence, the condition number of $\nabla^2_{\mathcal{M}_r} \ell(W)$ is approximately bounded by the condition number of $\nabla^2 \ell(W)$. Therefore, the convergence rate of RGD is mainly influenced by the condition number of $\nabla^2 \ell(W)$ only. In contrast, as shown in (5), the convergence rate of factorization-based algorithms depends not only on the condition number of $\nabla^2 \ell(W)$ but also on that of $\mathcal{J}_{\mathcal{G}}(P, Q)$, which is very large if the condition number of $W$ is very large.

- *Designing Riemannian Metrics for Improved Convergence.* The geometric structure of the Riemannian manifold allows us to flexibly design a Riemannian metric to improve the convergence of RGD. As previously discussed, the condition number of $\nabla^2_{\mathcal{M}_r}\ell(W)$ is approximately bounded by the condition number of $\nabla^2\ell(W)$. Therefore, it suffices to construct a metric on $\mathbb{T}_W$ such that, when extended to $\mathbb{E}$, the condition number of $\nabla^2\ell(W)$ is reduced. Many classical optimization algorithms, such as AdaGrad and AdamW, inherently define a metric on $\mathbb{E}$ that reduces the condition number of $\nabla^2\ell(W)$ by applying different learning rates to different components of the gradient. Naturally, we can restrict such a metric on $\mathbb{E}$ to $\mathbb{T}_W$ to obtain a Riemannian metric that reduces the condition number of $\nabla^2_{\mathcal{M}_r}\ell(W)$, resulting in accelerated RGD algorithms. This approach leads to two new RGD algorithms: RAdamW and RAdaGrad, which are the main contributions of this paper.

# 3 The Proposed Algorithms

For simplicity, we present our algorithms with $K = 1$. In this case, $W \in \mathbb{E} = \mathbb{R}^{n_0 \times n_1}$ and $\mathcal{M}_r = \mathcal{M}_{r_1}$. For convenience, we set $r = r_1$. Additionally, we omit the superscript $(k)$ throughout to simplify the notation. Extending the case $K = 1$ to a general $K$ is straightforward.

Recall that the Riemannian Gradient Descent (RGD) algorithm can be written as:

$$W_{t+1} = \mathcal{R}(W_t - \gamma_t \nabla_{\mathcal{M}_r}\ell(W_t)), \tag{11}$$

where $\nabla_{\mathcal{M}_r}\ell(W_t)$ is the Riemannian gradient, which depends on the Riemannian metric of $\mathcal{M}_r$, and $\mathcal{R}$ is a retraction operator.

To develop practical and efficient RGD algorithms, we must select both a Riemannian metric and a retraction operator $\mathcal{R}$ in (11). For all algorithms, we choose the retraction operator $\mathcal{R}$ to be the $r$-truncated singular value decomposition (SVD), i.e., $\mathcal{R} = \mathcal{H}_r$, where $\mathcal{H}_r : \mathbb{E} \to \mathcal{M}_r$ is defined as $\mathcal{H}_r(Z) = \sum_{i=1}^{r} \sigma_i u_i v_i^\top$, given the SVD of $Z = \sum_i \sigma_i u_i v_i^\top$.

For the Riemannian metric, we either adopt the standard metric, resulting in the Plain RGD algorithm presented in Section 3.1, or use a modified metric inspired by AdaGrad and AdamW, leading to the RAdaGrad and RAdamW algorithms described in Sections 3.2 and 3.3, respectively.

## 3.1 Plain RGD — Riemannian Gradient Descent with the Standard Metric

We choose $\mathcal{R} = \mathcal{H}_r$ and equip $\mathcal{M}_r$ with the standard metric from the Euclidean space $\mathbb{E}$. Under this setting, the Riemannian gradient is given by $\nabla_{\mathcal{M}_r}\ell(W) = \mathcal{P}^{(e)}_{\mathbb{T}_W}\nabla_e\ell(W)$, where $\mathcal{P}^{(e)}_{\mathbb{T}_W}$ denotes the standard orthogonal projection onto the tangent space $\mathbb{T}_W$, and $\nabla_e\ell(W)$ represents the standard gradient of in $\mathbb{E}$ .

With this setup, the Riemannian Gradient Descent (RGD) from (11) becomes:

$$W_{t+1} = \mathcal{H}_r(W_t - \gamma \mathcal{P}^{(e)}_{\mathbb{T}_t}\nabla_e\ell(W_t)), \tag{12}$$

where, for simplicity, $\mathbb{T}_t = \mathbb{T}_{W_t}$. We refer to the algorithm in (12) as the plain RGD. This algorithm serves as a baseline for the RGD method, and we use it to demonstrate the RGD method in the simplest setting.

Throughout the computation, $W_t$ is represented by its SVD $W_t = U_t \Sigma_t V_t^\top$, where $\Sigma_t \in \mathbb{R}^{r \times r}$, $U_t \in \mathbb{R}^{n_0 \times r}$, and $V_t \in \mathbb{R}^{n_1 \times r}$ are matrices containing the singular values and singular vectors, respectively. Below, we list the computations involved in the plain RGD algorithm (12):

- Since $G_t := \nabla_e\ell(W_t)$ is the standard gradient in $\mathbb{E}$, it can be directly obtained using standard backpropagation.
- With the help of $U_t$ and $V_t$, the tangent space $\mathbb{T}_t$ is expressed as:

$$\mathbb{T}_t = \left\{ U_t X^\top + Y V_t^\top \mid X \in \mathbb{R}^{n_1 \times r},\ Y \in \mathbb{R}^{n_0 \times r} \right\}. \tag{13}$$

An orthogonal basis for $\mathbb{T}_t$ can be formed, under which the projection $\mathcal{P}^{(e)}_{\mathbb{T}_t}(G_t)$ is computed as:

$$\mathcal{P}^{(e)}_{\mathbb{T}_t}(G_t) = U_t U_t^\top G_t + G_t V_t V_t^\top - U_t U_t^\top G_t V_t V_t^\top. \tag{14}$$

Since $Z_t := W_t - \mathcal{P}_{\mathbb{T}_t}^{(e)}(G_t) \in \mathbb{T}_t$ is expressed in the form of (13), we only need to compute the coefficient matrices $U_t^\top G_t$, $G_t V_t$, and $U_t^\top G_t V_t$ in the computation of $\mathcal{P}_{\mathbb{T}_t}^{(e)}(G_t)$. These computations can be efficiently performed using $\mathcal{O}(r)$ matrix-vector products with $G_t$.

- Since $Z_t \in \mathbb{T}_t$ and every matrix in $\mathbb{T}_t$ has a rank at most $2r$, the operator $\mathcal{H}_r(Z_t)$ finds the best rank-$r$ approximation of a rank-$2r$ matrix. This computation involves two QR decompositions of size $n_0 \times 2r$ and $n_1 \times 2r$, one matrix-matrix product of size $2r \times 2r$, and one SVD of size $2r \times 2r$. The total computational cost for these operations is $\mathcal{O}((n_0 + n_1)r^2 + r^3)$.

In summary, the computation in (12), in addition to evaluating $\nabla_e \ell(W_t)$, involves $\mathcal{O}(r)$ matrix-vector products and an additional $\mathcal{O}((n_0 + n_1)r^2 + r^3)$ operations. The memory complexity is $\mathcal{O}((n_0 + n_1)r)$. These complexities are of the same order as those of factorization-based algorithms. More details are provided in Appendix A.

## 3.2 RAdaGrad — Riemannian Gradient Descent with an Adaptive Data-Driven Metric

RGD can be accelerated by choosing an appropriate adaptive metric on the Riemannian manifold $\mathcal{M}_r$. Specifically, we can define new weighted inner products $\langle X, Y \rangle_{g_t} = \langle X, \mathcal{A}_t Y \rangle$ for any $X, Y \in \mathbb{T}_t$ on each tangent space $\mathbb{T}_t$ of the Riemannian manifold, where $\mathcal{A}_t : \mathbb{T}_t \to \mathbb{T}_t$ is a linear operator serving as weights for the inner product.

As we have previously argued, the new metric should be able to reduce the condition number of $\nabla^2_{\mathcal{M}_r} \ell(W_t)$, which ultimately translates to a reduction in the condition number of $\nabla^2 \ell(W_t)$ under the metric on $\mathbb{T}_t$ extended to $\mathbb{E}$. With the low-rank constraint, PRGD [3] also demonstrates the effectiveness of defining a new metric on $\mathbb{T}_t$. When the low-rank constraint is absent, many classical optimization algorithms, such as AdaGrad[12], AdamW [26, 29] and Shampoo [16], inherently define a metric on $\mathbb{E}$ that reduces the condition number of $\nabla^2 \ell(W_t)$ by applying different learning rates to different components of the gradient. We modify and restrict such a metric on $\mathbb{E}$ to $\mathbb{T}_t$ to obtain a Riemannian metric that reduces the condition number of $\nabla^2_{\mathcal{M}_r} \ell(W_t)$.

At step $t$, any metric in $\mathbb{E}$ is in the form of

$$\langle X, Y \rangle_{g_t} = \langle X, \mathcal{A}_t Y \rangle, \quad \forall \, X, Y \in \mathbb{E}, \tag{15}$$

where $\mathcal{A}_t : \mathbb{E} \to \mathbb{E}$ is a linear operator serving as weights for the inner product. Ideally, the best choice for $\mathcal{A}_t$ would be a multiple of $\nabla^2_e \ell(W_t)$, the Euclidean Hessian of $\ell$ under the standard metric in $\mathbb{E}$. However, computing $\nabla^2_e \ell(W_t)$ is very expensive. A common approach [9], [31, Chapter 10, 11, and 12] is to use the accumulation of the outer products of $G_t := \nabla_e \ell(W_t)$, i.e., we choose $\mathcal{A}_t = (\sum_t \langle G_t, \cdot \rangle G_t)^{1/2}$. Nevertheless, computing the gradient under this metric involves the inverse of $\mathcal{A}_t$, which is computationally expensive and may even be infeasible. To address this issue, we further approximate $\mathcal{A}_t$ using several efficient methods:

- In AdaGrad [12] and AdamW [26, 29], we use a diagonal operator to approximate $\mathcal{A}_t$, yielding

$$\mathcal{A}_t Y = A_t \circ Y, \qquad [A_t]_{ij} = \left( \varepsilon + \sum_t [G_t]_{ij}^2 \right)^{1/2}, \text{ for } i = 1, \dots, n_0, \ j = 1, \dots, n_1. \tag{16}$$

Here, $\circ$ denotes entry-wise multiplication, and $\varepsilon$ is a small positive number for numerical stability. The inverse of $\mathcal{A}_t$ is simply the entry-wise multiplication by the inverse of each entry of $A_t$.

- In Shampoo [16], we exploit the matrix structure of the variables by using the Kronecker product to approximate $\mathcal{A}_t$, which gives

$$\mathcal{A}_t Y = L_t^{\frac{1}{4}} Y R_t^{\frac{1}{4}}, \quad \text{where} \quad L_t = \varepsilon I + \sum_t G_t G_t^\top, \quad R_t = \varepsilon I + \sum_t G_t^\top G_t. \tag{17}$$

Here again, $\varepsilon$ is a small positive number for numerical stability. The inverse of $\mathcal{A}_t$ is simply $\mathcal{A}_t^{-1} Y = L_t^{-\frac{1}{4}} Y R_t^{-\frac{1}{4}}$.

We aim to utilize both the matrix structure of the variable $W$ and the easy invertibility of diagonal approximations. Thus, we combine both approximations and use the following $\mathcal{A}_t$, which is both diagonal and in Kronecker product form:

$$\mathcal{A}_t Y = L_t^{\frac{1}{4}} Y R_t^{\frac{1}{4}}, \quad \text{where } L_t = \varepsilon I + \sum_t \text{diag}(G_t G_t^\top), \quad R_t = \varepsilon I + \sum_t \text{diag}(G_t^\top G_t). \tag{18}$$

With this new metric on $\mathbb{E}$, the condition number of the Hessian $\nabla^2 \ell(W_t)$ is reduced. Then, we restrict this metric onto $\mathbb{T}_t$ to obtain our Riemannian metric, under which the condition number of $\nabla^2_{\mathcal{M}_r} \ell(W_t)$ is small since it is approximately bounded by that of $\nabla^2 \ell(W_t)$. More explicitly, the Riemannian metric we used here is: For any $X, Y \in \mathbb{T}_t$, we define their inner product through (15) with $\mathcal{A}_t$ from (17).

Under this new Riemannian metric, the Riemannian gradient is

$$\nabla^{(g)}_{\mathcal{M}_r} \ell(W_t) = \mathcal{P}^{(g)}_{\mathbb{T}_t}\big(\nabla_g \ell(W_t)\big) = \mathcal{P}^{(g)}_{\mathbb{T}_t}(\mathcal{A}_t^{-1} \nabla_e \ell(W_t)) = \mathcal{P}^{(g)}_{\mathbb{T}_t}(L_t^{-\frac{1}{4}} G_t R_t^{-\frac{1}{4}}), \qquad (19)$$

where $\mathcal{P}^{(g)}_{\mathbb{T}_t}$ denotes the orthogonal projection onto the tangent space $\mathbb{T}_t$ under the weighted inner product in (15) and (18). We choose $\mathcal{R} = \mathcal{H}_r$ and apply the general RGD framework in (11). This gives

$$(\textbf{RAdaGrad}) \begin{cases} G_t = \nabla_e \ell(W_t), \\ L_t = \beta_1 L_{t-1} + (1 - \beta_1)\mathrm{diag}(G_t G_t^\top), \\ R_t = \beta_2 R_{t-1} + (1 - \beta_2)\mathrm{diag}(G_t^\top G_t), \\ W_{t+1} = \mathcal{H}_r\left(W_t - \gamma_t \mathcal{P}^{(g)}_{\mathbb{T}_t}\big(L_t^{-\frac{1}{4}} G_t R_t^{-\frac{1}{4}}\big)\right). \end{cases} \qquad (20)$$

Here, we introduce the parameters $\beta_1$ and $\beta_2$ to use a weighted accumulation instead of direct accumulation in (17) and AdaGrad, providing more flexibility. The proposed algorithm is referred to as RAdaGrad.

Similar to the plain RGD, throughout the computation of RAdaGrad, $W_t$ is represented as its SVD $W_t = U_t \Sigma_t V_t^\top$. Below, we list the computation in RAdaGrad (20):

- $G_t$ is computed using standard backpropagation, the same as in plain RGD.

- $L_t$ and $R_t$ are diagonal matrices whose diagonals are actually the squared row norms and column norms of $G_t$, respectively. Thus, it requires only a computational complexity of $\mathcal{O}(n_1 n_2)$ or even lower if $G_t$ carries some structure.

- Since the metric weighting operator $\mathcal{A}_t$ is in Kronecker product form, the column vector space $\mathbb{R}^{n_0}$ and the row vector space $\mathbb{R}^{n_1}$ are weighted separately by $L_t^{\frac{1}{4}}$ and $R_t^{\frac{1}{4}}$ respectively. To find an orthogonal basis of $\mathbb{T}_t$, we only need to orthogonalize $U_t$ under $L_t^{\frac{1}{4}}$-weighted inner product to obtain $\tilde{U}_t = U_t(U_T^\top L^{\frac{1}{4}} U)^{-\frac{1}{2}}$ and $V_t$ under $R_t^{\frac{1}{4}}$-weighted inner product to obtain $\tilde{V}_t = V_t(V_t^\top R^{\frac{1}{4}} V)^{-\frac{1}{2}}$. Similar to (14), the projection $\mathcal{P}^{(g)}_{\mathbb{T}_t}$ is given by:

$$\mathcal{P}^{(g)}_{\mathbb{T}_t}(L_t^{-\frac{1}{4}} G_t R_t^{-\frac{1}{4}}) = \widetilde{U}_t \widetilde{U}_t^\top G_t R_t^{-\frac{1}{4}} + L_t^{-\frac{1}{4}} G_t \widetilde{V}_t \widetilde{V}_t^\top - \widetilde{U}_t \widetilde{U}_t^\top G_t \widetilde{V}_t \widetilde{V}_t^\top, \qquad (21)$$

Again, we only need to calculate the coefficient matrices $\widetilde{U}_t^\top G_t R_t^{-\frac{1}{4}}$, $L_t^{-\frac{1}{4}} G_t \widetilde{V}_t$, and $\widetilde{U}_t^\top G_t \widetilde{V}_t$ in the computation of $\mathcal{P}^{(g)}_{\mathbb{T}_t}(L_t^{-\frac{1}{4}} G_t R_t^{-\frac{1}{4}})$. These computations can be efficiently performed using $\mathcal{O}(r)$ matrix-vector products with $G_t$.

- The computation of $\mathcal{H}_r(Z_t)$ is the same as in plain RGD, with a total computational cost of $\mathcal{O}((n_0 + n_1)r^2 + r^3)$.

In total, the computation of RAdaGrad is, in addition to evaluating $G_t$, $\mathcal{O}(r)$ matrix-vector products with $G_t$ and an extra $\mathcal{O}((n_0 + n_1)r^2 + r^3)$ operation. The memory usage is $O((n_0 + n_1)r)$. Both complexities are in the same order as factorization-based algorithms. The detailed calculations can be found in the Appendix B.

### 3.3 RAdamW — RGD with an Adaptive Momentum and Weight Decay

Momentum is a commonly used acceleration technique for gradient-based algorithms. Adam has demonstrated certain advantages on convex problems and the Stiefel manifold [2, 36, 41]. To further enhance the performance of RAdaGrad, we introduce the momentum from AdamW [26, 29] into the RAdaGrad algorithm.

In fact, RAdamW can be regarded as a variant of the Riemannian Conjugate Gradient (RCG) algorithm. The resulting RAdamW algorithm is as follows

$$
(\textbf{RAdamW}) \begin{cases} G_t = \nabla_e \ell(W_t), \\ L_t = \beta_1 L_{t-1} + (1-\beta_1)\text{diag}(G_t G_t^\top), \\ R_t = \beta_2 R_{t-1} + (1-\beta_2)\text{diag}(G_t^\top G_t), \\ M_t = \beta_3 M_{t-1} + (1-\beta_3)G_t, \\ W_{t+1} = \mathcal{H}_r\left((1-\theta)W_t - \gamma_t \mathcal{P}_{\mathbb{T}_t}^{(g)}\left(L_t^{-\frac{1}{4}} M_t R_t^{-\frac{1}{4}}\right)\right). \end{cases}
\tag{22}
$$

where $M_t$ represents the update direction at $t$ step, which is a linear combination of the original gradient $G_t$ and $M_{t-1}$. Its detailed computation is almost same with RAdaGrad. Here we omit the detail.

## 4 Experiments

To comprehensively evaluate the performance of the RAdaGrad and RAdamW algorithms, we conduct experiments across a range of tasks, including fine-tuning large language models (GPT-2 [33]), fine-tuning diffusion models (Mix-of-Show [14] and Stable Diffusion V1.5 [35]), and deep neural network (DNN) compression tasks.

For the large model fine-tuning, we compare two kinds of optimization algorithms: SGD-based algorithms and AdamW-based algorithms. The SGD-based algorithms include LoRA using SGD (denoted as SGD) [21], ScaledGD [40, 47], plain RGD [45], and RAdaGrad (Algorithm 20). This comparison allows for an evaluation of the improvements brought by RAdaGrad over conventional SGD approaches. On the other hand, the AdamW-based algorithms include AdamW (vanilla LoRA) [21], ScaledAdamW [40, 47], and RAdamW (Algorithm 31). By comparing these AdamW-based algorithms, we assess the performance advantages of RAdamW.

We validate the effectiveness of the proposed algorithms in DNNs compression on the MNIST and CIFAR10 datasets using fully connected and convolutional networks. The experimental results (detailed in Appendix E) demonstrate that RAdaGrad and RAdamW, as plug-and-play optimizers, can be seamlessly integrated into various network architectures. The large model fine-tuning experiments are conducted on the NVIDIA A100 GPUs, while the DNN compression experiments are performed on a system equipped with an AMD Ryzen 7 7800X3D 8-core CPU and an Nvidia RTX 4090 GPU. The code for all our experiments can be found in the supplementary materials.

### 4.1 Large Models Fine-Tuning

In this section, we evaluate RAdaGrad and RAdamW algorithms on large model fine-tuning. We present the experimental results for these algorithms on the fine-tuning of diffusion models (Mix-of-Show, see sections 4.1.1 and C.3) and Stable Diffusion,see Appendix C.2) and large language models (GPT-2, see sections 4.1.2 and C.1).

### 4.1.1 Mix-of-Show

This section preforms our experiment details on Mix-of-show model [14]. We use OpenAI's clip-vit-base-patch16 model to calculate the CLIP score, measuring the consistency between generated images and text prompts. Frèchet Inception Distance (FID) is a commonly used metric for evaluating class-conditioned generative models. The specific parameter settings for each algorithm are provided in Table 8 in Appendix C.3. During the experiments, we employ a linear learning rate schedule and set the rank $r = 1, 4$. The CLIP and FID scores are reported in Table 1. From Table 1, it is evident that RAdaGrad and RAdamW significantly outperform other algorithms, whether they are SGD-based or AdamW-based. Specifically, when $r = 1$ or $4$ ,RAdamW significantly outperforms Scaled AdamW, achieving nearly 1 point higher CLIP score and about 10 points lower FID score, demonstrating the effectiveness of our proposed Riemannian algorithms.

### 4.1.2 GPT-2

We evaluate the performance of RAdaGrad and RAdamW optimizers on GPT-2. To ensure fairness, all hyperparameters except for the learning rate and weight decay are kept consistent, and the optimal

Table 1: Experiments for EDLoRA on Clip and FID score.

| Methods | Rank | CLIP↑ | FID↓ | Rank | CLIP↑ | FID↓ |
|---|---|---|---|---|---|---|
| | | Metrics | | | Metrics | |
| SGD | 1 | 22.93 | 68.43 | 4 | 27.88 | 66.57 |
| Scaled GD | 1 | 32.19 | 53.75 | 4 | 32.14 | 48.65 |
| plain RGD | 1 | 32.56 | 69.39 | 4 | 32.94 | 52.07 |
| **RAdaGrad(ours)** | 1 | **32.84** | **50.45** | 4 | **33.95** | **47.01** |
| AdamW | 1 | 32.84 | 53.11 | 4 | 33.75 | 52.16 |
| Scaled AdamW | 1 | 31.94 | 66.77 | 4 | 33.61 | 60.71 |
| **RAdamW(ours)** | 1 | **33.02** | **50.84** | 4 | **34.69** | **51.14** |

combinations are determined via grid search (complete implementation details and hyperparameter settings can be found in Appendix C.1). On the E2E natural language generation benchmark [23], we fine-tune GPT-2 small with ranks of 4, 8, and 16. The results are summarized in Table 2.

The experiments demonstrate that RAdaGrad and RAdamW consistently outperform other methods across all evaluation metrics, validating the efficiency and accuracy of our proposed framework in LLMs fine-tuning. Moreover, all RGD-based algorithms generally outperform all other algorithms, further demonstrating the advantages of the RGD framework in effectively leveraging the smooth manifold structure of low-rank matrices and adaptive metrics. Notably, RAdaGrad achieves performance improvements with reduced memory compared to AdamW-based methods, as it avoids storing historical $G$ matrices. Meanwhile, RAdamW surpassed both AdamW and scaled AdamW across all evaluation metrics. This performance gain can be attributed to its combination of Riemannian geometry-aware updates and adaptive learning rates based on first- and second-moment estimation.

Table 2: Scores for LoRA fine-tuning ($r = 4, 8$ and 16) of GPT-2 on E2E Natural Language Generation challenge.

| rank | Method | BLEU | NIST | MET | ROUGE-L | CIDEr |
|---|---|---|---|---|---|---|
| | | | | E2E | | |
| 4 | SGD | 66.6 | 8.54 | 44.2 | 68.2 | 2.32 |
| | Scaled GD | 68.5 | 8.72 | 45.5 | 69.4 | 2.40 |
| | plain RGD | 69.2 | 8.77 | 45.7 | 70.1 | 2.45 |
| | **RAdaGrad (ours)** | **69.8** | **8.80** | **46.5** | **71.1** | **2.49** |
| | AdamW | 69.1 | 8.75 | 46.0 | 70.5 | 2.47 |
| | Scaled AdamW | 69.5 | 8.80 | 46.2 | 70.9 | 2.48 |
| | **RAdamW (ours)** | **69.8** | **8.81** | **46.5** | **71.1** | **2.51** |
| 8 | SGD | 65.8 | 8.46 | 43.5 | 68.7 | 2.33 |
| | Scaled GD | 68.8 | 8.75 | 45.3 | 69.4 | 2.43 |
| | plain RGD | 69.1 | 8.73 | 46.0 | 70.7 | 2.45 |
| | **RAdaGrad (ours)** | **70.1** | **8.80** | **46.8** | **71.7** | **2.51** |
| | AdamW | 69.4 | 8.77 | 46.2 | 71.0 | 2.46 |
| | Scaled AdamW | 69.7 | 8.80 | **46.5** | 71.3 | 2.52 |
| | **RAdamW (ours)** | **70.3** | **8.82** | **46.5** | **71.8** | **2.53** |
| 16 | SGD | 65.4 | 8.07 | 40.7 | 67.0 | 2.07 |
| | Scaled GD | 68.8 | 8.75 | 45.0 | 69.2 | 2.39 |
| | plain RGD | 68.7 | 8.67 | 46.1 | 70.8 | 2.44 |
| | **RAdaGrad (ours)** | **70.3** | **8.84** | **46.6** | **71.8** | **2.52** |
| | AdamW | 69.5 | 8.77 | 46.4 | 71.2 | 2.48 |
| | Scaled AdamW | 69.8 | 8.79 | 46.5 | **71.7** | 2.51 |
| | **RAdamW (ours)** | **70.1** | **8.85** | **46.6** | 71.6 | **2.52** |

## 5  Conclusion

Most existing algorithms for finding low-rank weights in large models are based on matrix factorization and are constrained by the condition numbers of the Jacobian and Hessian operators. To address these limitations, we propose a general RGD framework for efficiently finding low-rank

matrix weights in DNNs. This framework avoids the redundant representation introduced by matrix factorization, avoids the influence of the Jacobian condition number, and mitigates the impact of the Hessian condition number by selecting appropriate metrics. Inspired by AdaGrad and AdamW, we extend their metrics to Riemannian manifolds and propose two new algorithms: RAdaGrad and RAdamW. These algorithms completely avoid the influence of the Jacobian condition number and significantly alleviate the effect of the Hessian condition number.

Through experiments on large model fine-tuning and DNN compression tasks, our plug-and-play optimizers are shown to outperform state-of-the-art methods and are applicable to various network architectures. Our work not only introduces RAdaGrad and RAdamW but also establishes a unified framework that provides a novel method to finding the low-rank weights and offers a fresh perspective for designing efficient algorithms. In the future, we plan to extend this framework to address the problem of finding low-rank tensor weights in DNNs.

**Limitations.** The rank-$r$ Riemannian manifold constraint in low-rank adaptation methods inherently limits their expressiveness compared to full-parameter fine-tuning.

**Broader Impact.** The proposed RAdaGrad and RAdamW optimizers demonstrate broad compatibility across diverse DNN architectures, which is useful to fields like healthcare, farming, and schools, especially in places with limited resources.

## Acknowledgments and Disclosure of Funding

The authors would like to thank the anonymous referees very much for their careful reading and valuable comments, which greatly improved the quality of this manuscript. Jian-Feng Cai is partially supported by Hong Kong Research Grant Council GRF 16307023, GRF 16306124, and GRF 16307325.

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

# A  Computational Details in the Plain RGD

In this section, we provide the computational details for plain RGD (see Eq. (12) in Section 3.1 of the main text). Recall that plain RGD generates the iterates via

$$W_{t+1} = \mathcal{H}_r \left( W_t - \gamma \mathcal{P}_{\mathbb{T}_t}^{(e)} \nabla_e \ell(W_t) \right), \tag{23}$$

where $\mathcal{P}_{\mathbb{T}_t}^{(e)}$ is the orthogonal projection from $\mathbb{E}$ onto the tangent space $\mathbb{T}_t$, and $\nabla_e \ell(W_t)$ denotes the standard gradient of the loss function $\ell$ in $\mathbb{E}$.

Throughout the computation, the rank-$r$ weight matrix $W_t$ is represented by its singular value decomposition (SVD) as $W_t = U_t \Sigma_t V_t^\top$, where $\Sigma_t \in \mathbb{R}^{r \times r}$ is a diagonal matrix whose diagonal elements are the singular values of $W_t$, $U_t \in \mathbb{R}^{n_0 \times r}$ contains the left singular vectors, and $V_t \in \mathbb{R}^{n_1 \times r}$ contains the right singular vectors.

- *Computation of $G_t := \nabla_e \ell(W_t)$.* Since $G_t$ is the standard Euclidean gradient of the loss function $\ell$ in $\mathbb{E}$, it can be computed efficiently via standard backpropagation.

- *Computation of $Z_t := W_t - \gamma \mathcal{P}_{\mathbb{T}_t}^{(e)}(G_t)$.* We first establish representations for the matrices involved in computing $Z_t$.

  - **Representation of $\mathcal{P}_{\mathbb{T}_t}^{(e)}(G_t)$ in terms of $U_t$, $V_t$, and $G_t$.** Let $V_t^\perp \in \mathbb{R}^{(n_1-r) \times n_1}$ and $U_t^\perp \in \mathbb{R}^{n_0 \times (n_1-r)}$ be the orthogonal complements of $V_t$ and $U_t$, respectively. The tangent space $\mathbb{T}_t$ admits the decomposition:

  $$\mathbb{T}_t = \left\{ U_t X^\top + Y V_t^\top \mid X \in \mathbb{R}^{n_1 \times r}, \ Y \in \mathbb{R}^{n_0 \times r} \right\},$$
  $$= \left\{ U_t K_1 V_t^\top \mid K_1 \in \mathbb{R}^{r \times r} \right\} \oplus \left\{ U_t^\perp K_2 V_t^\top \mid K_2 \in \mathbb{R}^{(n_0-r) \times r} \right\}$$
  $$\oplus \left\{ U_t K_3 (V_t^\perp)^\top \mid K_3 \in \mathbb{R}^{r \times (n_1-r)} \right\}.$$

  Using this decomposition, the projection $\mathcal{P}_{\mathbb{T}_t}^{(e)}(G_t)$ evaluates to:

  $$\begin{aligned}
  \mathcal{P}_{\mathbb{T}_t}^{(e)}(G_t) &= U_t U_t^\top G_t V_t V_t^\top + + U_t U_t^\top G_t V_t^\perp (V_t^\perp)^\top + U_t^\perp (U_t^\perp)^\top G_t V_t V_t^\top \\
  &= U_t U_t^\top G_t V_t V_t^\top + U_t U_t^\top G_t (I - V_t V_t^\top) + (I - U_t U_t^\top) G_t V_t V_t^\top.
  \end{aligned} \tag{24}$$

  - **Expression of $Z_t$ in terms of $U_t$, $V_t$, $\Sigma_t$, and $G_t$.** Using the projection, we can reformulate $Z_t$ as

  $$\begin{aligned}
  Z_t &= U_t \Sigma_t V_t^\top - \gamma U_t U_t^\top G_t V_t V_t^\top - \gamma U_t U_t^\top G_t (I - V_t V_t^\top) - \gamma (I - U_t U_t^\top) G_t V_t V_t^\top \\
  &= U_t \underbrace{\left( \Sigma_t - \gamma U_t^\top G_t V_t \right)}_{Y_0} V_t^\top - \gamma U_t \underbrace{U_t^\top G_t (I - V_t V_t^\top)}_{Y_1^\top} - \gamma \underbrace{(I - U_t U_t^\top) G_t V_t}_{Y_2} V_t^\top \\
  &:= U_t Y_0 V_t^\top - \gamma U_t Y_1^\top - \gamma Y_2 V_t^\top.
  \end{aligned} \tag{25}$$

Therefore, to compute $Z_t$, we only need to compute the coefficient matrices $Y_0 \in \mathbb{R}^{r \times r}$, $Y_1 \in \mathbb{R}^{n_1 \times r}$, and $Y_2 \in \mathbb{R}^{n_0 \times r}$. The detailed procedure is as follows:

  - Compute $G_t V_t$, $G_t^T U_t$, and $U_t^\top (G_t V_t)$.
  - Compute $Y_0 = \Sigma_t - \gamma (U_t^\top G_t V_t)$.
  - Compute $Y_1 = (G_t^T U_t) - V_t (U_t^\top G_t V_t)^\top$ and $Y_2 = (G_t V_t) - U_t (U_t^\top G_t V_t)$.

This procedure requires $\mathcal{O}(n_0 n_1 r)$ operations and $\mathcal{O}((n_0 + n_1)r)$ memory.

- *Computation of $\mathcal{H}_r(Z_t)$.* The rank of $Z_t$ is at most $2r$ due to its structured representation:

$$Z_t = \underbrace{\left[ \begin{array}{cc} U_t & Y_2 \end{array} \right]}_{n_0 \times 2r} \underbrace{\left[ \begin{array}{cc} Y_0 & -\gamma I \\ -\gamma I & 0 \end{array} \right]}_{2r \times 2r} \underbrace{\left[ \begin{array}{c} V_t^\top \\ Y_1^\top \end{array} \right]}_{2r \times n_1},$$

Since $U_t^\top Y_2 = 0$, the QR decomposition $Y_2 = Q_2 R_2$ yields an orthognal matrix $[U_t \ Q_2] \in \mathbb{R}^{n_0 \times 2r}$. Similarly, the QR decomposition $Y_1 = Q_1 R_1$ gives an orthognal matrix $[V_t \ Q_1] \in$

$\mathbb{R}^{n_1 \times 2r}$. This allows us to rewrite $Z_t$ as:

$$Z_t = \begin{bmatrix} U_t & Q_2 \end{bmatrix} \underbrace{\begin{bmatrix} Y_0 & -\gamma R_1^\top \\ -\gamma R_2 & 0 \end{bmatrix}}_{2r \times 2r} \begin{bmatrix} V_t^\top \\ Q_1^\top \end{bmatrix}, \qquad (26)$$

where both the left and right factors are orthogonal matrices. Therefore, computing the SVD of $Z_t$ reduces to finding the SVD of the central $2r \times 2r$ matrix. By retaining only the leading $r$ singular values and corresponding singular vectors, we obtain the rank-$r$ approximation $W_{t+1} := \mathcal{H}_r(Z_t)$ in its SVD form. The complete computational procedure consists of the following steps:

- Compute QR decompositions $Y_1 = Q_1 R_1$ and $Y_2 = Q_2 R_2$.
- Compute the SVD of

$$\begin{bmatrix} Y_0 & -\gamma R_1^\top \\ -\gamma R_2 & 0 \end{bmatrix} = C \Lambda D^T$$

- Compute

$$U_{t+1} = [U_t \, Q_2] \cdot C(:, 1:r), \quad \Sigma_{t+1} = \Lambda(1:r, 1:r), \quad V_{t+1} = [V_t \, Q_1] \cdot D(:, 1:r).$$

The whole procedure requires two QR decompositions, one SVD of size $2r \times 2r$, and two matrix-matrix products. The computational complexity is $\mathcal{O}((n_0 + n_1)r^2)$, and the memory used is $\mathcal{O}((n_0 + n_1)r)$.

## B   Computational Details in RAdaGrad and RAdamW

In this section, we provide the computational details of the Riemannian Gradient Descent with an Adaptive Data-Driven Metric (RAdaGrad, see Equation (20) in Section 3.2 of the main text). Recall that the standard RAdaGrad generates updates using the following formula:

$$\textbf{(RAdaGrad)} \begin{cases} G_t = \nabla_e \ell(W_t), \\ L_t = \beta_1 L_{t-1} + (1 - \beta_1) \text{diag}(G_t G_t^\top), \\ R_t = \beta_2 R_{t-1} + (1 - \beta_2) \text{diag}(G_t^\top G_t), \\ W_{t+1} = \mathcal{H}_r \left( W_t - \gamma \mathcal{P}_{\mathbb{T}_t}^{(g)} \left( L_t^{-\frac{1}{4}} G_t R_t^{-\frac{1}{4}} \right) \right), \end{cases} \qquad (27)$$

where $\mathcal{P}_{\mathbb{T}_t}^{(g)}$ denotes the orthogonal projection from $\mathbb{E}$ to the tangent space $\mathbb{T}_t$ under the weighted inner product $\langle X, Y \rangle_{g_t} = \langle X, L^{\frac{1}{4}} Y R^{\frac{1}{4}} \rangle$ for $X, Y \in \mathbb{E}$, and $\nabla_e \ell(W_t)$ represents the standard gradient of the loss function $\ell$ in the Euclidean space $\mathbb{E}$.

We now present the computational details of the RAdaGrad algorithm. The matrix $W_t$ is expressed using its singular value decomposition (SVD) as $W_t = U_t \Sigma_t V_t^\top$, where $\Sigma_t \in \mathbb{R}^{r \times r}$ is a diagonal matrix whose diagonal elements are the singular values of $W_t$, and $U_t \in \mathbb{R}^{n_0 \times r}$ and $V_t \in \mathbb{R}^{n_1 \times r}$ are the left and right singular vector matrices, respectively. The computations involved in RAdaGrad are outlined below.

- *Computation of $G_t := \nabla_e \ell(W_t)$.* In RAdaGrad and RAdamW, $G_t$ is also the standard Euclidean gradient of the loss function $\ell$ in $\mathbb{E}$. Therefore, similar to the plain RGD, it can be directly obtained through backpropagation.

- *Computation of $L_t$ and $R_t$.* According to (27), $L_t$ and $R_t$ are diagonal matrices defined by the squared column 2-norms and squared row 2-norms of $G_t$, respectively. The computation involves the following steps:

  - Calculate the squared 2-norm of each column and each row of $G_t$.
  - Update the diagonal entries of $L_t$ and $R_t$ using the squared column norms and row norms, respectively.

  The total computational cost for these operations is $\mathcal{O}(n_0 n_1)$.

- *Computation of $Z_t := W_t - \gamma \mathcal{P}_{\mathbb{T}_t}^{(g)} (L_t^{-\frac{1}{4}} G_t R_t^{-\frac{1}{4}})$.* This is similar to the computation of $\mathcal{P}_{\mathbb{T}_t}^{(e)}(G_t)$ in plain RGD. We first establish representations for the matrices involved in computing $Z_t$.

- **Expression of** $\mathcal{P}^{(g)}_{\mathbb{T}_t}(L_t^{-\frac{1}{4}} G_t R_t^{-\frac{1}{4}})$. This is very similar to $\mathcal{P}^{(e)}_{\mathbb{T}_t}(G_t)$ in plain RGD. The only difference is that we need to find an orthogonal basis for $\mathbb{T}_t$ under the weighted inner products. Notice that the weighting operator in the weighted inner product $\langle \cdot, \cdot \rangle_{g_t}$ is in Kronecker product form and is therefore separable in the column vector and row vector spaces. Specifically, we orthogonalize $U_t$ using the $L_t^{\frac{1}{4}}$-inner product to obtain $\widetilde{U}_t = U_t(U_t^\top L_t^{\frac{1}{4}} U_t)^{-\frac{1}{2}}$, and orthogonalize $V_t$ using the $R_t^{\frac{1}{4}}$-inner product to obtain $\widetilde{V}_t = V_t(V_t^\top R_t^{\frac{1}{4}} V_t)^{-\frac{1}{2}}$. As a result, we have:

$$\mathcal{P}^{(g)}_{\mathbb{T}_t}(L_t^{-\frac{1}{4}} G_t R_t^{-\frac{1}{4}}) = \widetilde{U}_t \widetilde{U}_t^\top G_t R_t^{-\frac{1}{4}} + L_t^{-\frac{1}{4}} G_t \widetilde{V}_t \widetilde{V}_t^\top - \widetilde{U}_t \widetilde{U}_t^\top G_t \widetilde{V}_t \widetilde{V}_t^\top. \quad (28)$$

- **Representation of** $Z_t$ **in terms of** $U_t$, $V_t$, $\Sigma_t$, **and** $G_t$. Using the projection, we can reformulate $Z_t$ as:

$$\begin{aligned} Z_t &= W_t - \gamma \mathcal{P}^{(g)}_{\mathbb{T}_t}(L_t^{-\frac{1}{4}} G_t R_t^{-\frac{1}{4}}) \\ &= U_t \Sigma_t V_t - \gamma \widetilde{U}_t \widetilde{U}_t^\top G_t R_t^{-\frac{1}{4}} - \gamma L_t^{-\frac{1}{4}} G_t \widetilde{V}_t \widetilde{V}_t^\top + \gamma \widetilde{U}_t \widetilde{U}_t^\top G_t \widetilde{V}_t \widetilde{V}_t^\top. \end{aligned}$$

Each term can be expressed in terms of $U_t$, $V_t$, and $\Sigma_t$:

* $\widetilde{U}_t \widetilde{U}_t^\top G_t R_t^{-\frac{1}{4}} = U_t \underbrace{(U_t^\top L_t^{\frac{1}{4}} U_t)^{-1} U_t^\top}_{K_1^\top} G_t R_t^{-\frac{1}{4}} := U_t K_1^{\ \top} G_t R_t^{-\frac{1}{4}}.$

* $L_t^{-\frac{1}{4}} G_t \widetilde{V}_t \widetilde{V}_t^\top = L_t^{-\frac{1}{4}} G_t \underbrace{V_t (V_t^\top R_t^{\frac{1}{4}} V_t)^{-1}}_{K_2} V_t^\top := L_t^{-\frac{1}{4}} G_t K_2 V_t^\top.$

* $\widetilde{U}_t \widetilde{U}_t^\top G_t \widetilde{V}_t \widetilde{V}_t^\top = U_t K_1^\top G_t K_2 V_t^\top.$

Substituting these into the expression for $Z_t$, we obtain:

$$\begin{aligned} Z_t &= W_t - \gamma \mathcal{P}^{(g)}_{\mathbb{T}_t}(L_t^{-\frac{1}{4}} G_t R_t^{-\frac{1}{4}}) \\ &= U_t \Sigma_t V_t - \gamma U_t K_1^{\ \top} G_t R_t^{-\frac{1}{4}} - \gamma L_t^{-\frac{1}{4}} G_t K_2 V_t^\top + \gamma U_t K_1^{\ \top} G_t K_2 V_t^\top \\ &= U_t \underbrace{(\Sigma_t - \gamma K_1^\top G_t R_t^{-\frac{1}{4}} V_t - \gamma U_t^\top L_t^{-\frac{1}{4}} G_t K_2 + \gamma K_1^\top G_t K_2)}_{Y_0} V_t^\top \\ &\quad - \gamma U_t \underbrace{K_1^{\ \top} G_t R_t^{-\frac{1}{4}}(I - V_t V_t^\top)}_{Y_1^\top} - \gamma \underbrace{(I - U_t U_t^\top) L_t^{-\frac{1}{4}} G_t K_2}_{Y_2} V_t^\top \\ &:= U_t Y_0 V_t^\top - \gamma U_t Y_1^\top - \gamma Y_2 V_t^\top. \end{aligned} \quad (29)$$

Therefore, to compute $Z_t$, it is sufficient to calculate the coefficient matrices $Y_0 \in \mathbb{R}^{r \times r}$, $Y_1 \in \mathbb{R}^{n_1 \times r}$, and $Y_2 \in \mathbb{R}^{n_0 \times r}$. The detailed procedure is as follows:

- Compute $K_1 = U_t(U_t^\top L_t^{\frac{1}{4}} U_t)^{-1}$ and $K_2 = V_t(V_t^\top R_t^{\frac{1}{4}} V_t)^{-1}$.
- Compute $G_t^\top K_1$, $G_t K_2$, and $K_1^\top(G_t K_2)$.
- Compute $L_t^{-\frac{1}{4}}(G_t K_2)$ and $R_t^{-\frac{1}{4}}(G_t^\top K_1)$ by rescaling each row.
- Compute $Y_0 = \Sigma_t - \gamma\big((K_1^\top G_t R_t^{-\frac{1}{4}})V_t + U_t^\top(L_t^{-\frac{1}{4}} G_t K_2) - (K_1^\top G_t K_2)\big)$.
- Compute $Y_1 = (K_1^{\ \top} G_t R_t^{-\frac{1}{4}}) - \big((K_1^{\ \top} G_t R_t^{-\frac{1}{4}})V_t\big)V_t^\top$ and $Y_2 = (L_t^{-\frac{1}{4}} G_t K_2) - U_t\big(U_t^\top(L_t^{-\frac{1}{4}} G_t K_2)\big)$.

Since $L_t$ and $R_t$ are diagonal matrices, the multiplications involved in $L_t^{-\frac{1}{4}}$ and $R_t^{-\frac{1}{4}}$ rescale the rows and columns, making them computationally cheap. The entire procedure requires $\mathcal{O}(n_0 n_1 r)$ operations and $\mathcal{O}((n_0 + n_1)r)$ memory.

- *Computation of* $\mathcal{H}_r(Z_t)$. From (29), it follows that $Z_t$ can be rewritten in the following form:

$$Z_t = \underbrace{\begin{bmatrix} U_t & Y_2 \end{bmatrix}}_{n_0 \times 2r} \underbrace{\begin{bmatrix} Y_0 & -\gamma I \\ -\gamma I & 0 \end{bmatrix}}_{2r \times 2r} \underbrace{\begin{bmatrix} V_t^\top \\ Y_1^\top \end{bmatrix}}_{2r \times n_1} \quad (30)$$

with $U_t^\top Y_2 = 0$ and $V_t^\top Y_1 = 0$. This form is exactly the same as (26) in plain RGD, except for different $Y_0$, $Y_1$, and $Y_2$. Therefore, the computation of $W_{t+1} = \mathcal{H}_r(Z_t)$ is identical to that of plain RGD. As such, we omit the detailed computational steps here. Similar to plain RGD, the computational complexity is $\mathcal{O}((n_0 + n_1)r^2)$, and the memory usage is $\mathcal{O}((n_0 + n_1)r)$.

Regarding the computational details of the RAdamW algorithm (see Equation (22) in Section 3.3 of the main text), the RAdamW algorithm is as follows:

$$
(\textbf{RAdamW}) \begin{cases}
G_t = \nabla_e \ell(W_t), \\
L_t = \beta_1 L_{t-1} + (1 - \beta_1)\text{diag}(G_t G_t^\top), \\
R_t = \beta_2 R_{t-1} + (1 - \beta_2)\text{diag}(G_t^\top G_t), \\
M_t = \beta_3 M_{t-1} + (1 - \beta_3)G_t, \\
W_{t+1} = \mathcal{H}_r\left((1 - \theta)W_t - \gamma_t \mathcal{P}_{\mathbb{T}_t}^{(g)}\left(L_t^{-\frac{1}{4}} M_t R_t^{-\frac{1}{4}}\right)\right).
\end{cases} \tag{31}
$$

Note that compared to the RAdaGrad algorithm, the only difference is that $G_t$ in RAdaGrad is replaced with $M_t$ in the last line. As a result, the computational procedure remains almost the same. For this reason, we omit the computational details of the RAdamW algorithm here.

## C Supplementary Experiments for Large Models Fine-Tuning

### C.1 GPT-2

#### C.1.1 Sensitivity Analysis of Hyperparameters

To evaluate the stability of the proposed optimizer, we test the sensitivity of RAdaGrad and RAdamW to the parameters on GPT-2. Specifically, we validate their sensitivity to weight decay, learning rate, and smoothing parameters, with the results shown in Table 3, Table 4, and Table 5, respectively. The results indicate that all metrics exhibit slight fluctuations within the error range.

In detail, Table 3 presents the training loss and evaluation metrics under different weight decay settings. For these experiments, RAdaGrad's learning rate was fixed at 5e-3, RAdamW's learning rate was fixed at 8e-3, smoothing parameters were set to $\beta_1 = \beta_2 = 0.98$, and the rank was set to 4. Table 4 shows the training loss and evaluation metrics under varying learning rates, with weight decay fixed at 1e-2, smoothing parameters set to $\beta_3 = 0.9$, $\beta_1 = \beta_2 = 0.98$, and the rank set to 4. Table 5 provides the training loss and evaluation metrics for different smoothing parameters $\beta_1 = \beta_2$, with the learning rate fixed at 8e-3, smoothing parameter $\beta_3 = 0.9$, weight decay set to 1e-2, and the rank set to 4.

Table 3: Training loss and evaluation scores for the GPT-2 fine-tuned model ($r = 4$) under varying weight decay schedules.

| Methods | Weight Deacy | Training Loss | E2E | | | | |
| --- | --- | --- | --- | --- | --- | --- | --- |
| | | | BLEU | NIST | MET | ROUGE-L | CIDEr |
| RAdaGrad | 1e-2 | 2.56 | 69.8 | 8.80 | 46.5 | 71.1 | 2.49 |
| | 1e-3 | 2.56 | 70.0 | 8.82 | 46.5 | 71.2 | 2.51 |
| | 1e-4 | 2.56 | 69.7 | 8.79 | 46.5 | 70.9 | 2.50 |
| RAdamW | 1e-2 | 2.56 | 69.8 | 8.81 | 46.5 | 71.1 | 2.51 |
| | 1e-3 | 2.56 | 68.8 | 8.76 | 45.5 | 70.1 | 2.42 |
| | 1e-4 | 2.56 | 70.0 | 8.83 | 46.3 | 71.2 | 2.48 |

#### C.1.2 Training Loss Curve

To further demonstrate the advantages of RAdaGrad and RAdamW, we compare the loss within the same runtime, and the results are shown in Figure 1. These results highlight the effectiveness of the proposed optimizers.

Table 4: Training loss and evaluation scores for the GPT-2 fine-tuned model ($r = 4$) under varying learning rates.

| Methods | Learning Rate | Training Loss | E2E | | | | |
| | | | BLEU | NIST | MET | ROUGE-L | CIDEr |
|---|---|---|---|---|---|---|---|
| RAdaGrad | 1e-2 | 2.55 | 69.5 | 8.76 | 46.9 | 71.4 | 2.52 |
| | 5e-3 | 2.56 | 69.8 | 8.80 | 46.5 | 71.1 | 2.49 |
| | 1e-3 | 2.59 | 69.3 | 8.79 | 45.9 | 70.5 | 2.44 |
| RAdamW | 1e-2 | 2.57 | 69.2 | 8.76 | 46.0 | 70.7 | 2.46 |
| | 5e-3 | 2.56 | 69.1 | 8.75 | 46.1 | 70.7 | 2.47 |
| | 1e-3 | 2.59 | 69.2 | 8.76 | 45.8 | 70.2 | 2.44 |

Table 5: Training loss and evaluation scores for the GPT-2 fine-tuned model ($r = 4$) under varying smooth parameters.

| Methods | Parameter $\beta_1, \beta_2$ | Training Loss | E2E | | | | |
| | | | BLEU | NIST | MET | ROUGE-L | CIDEr |
|---|---|---|---|---|---|---|---|
| RAdamW | 0.96 | 2.56 | 69.3 | 8.81 | 45.9 | 70.5 | 2.45 |
| | 0.98 | 2.56 | 69.8 | 8.81 | 46.5 | 71.1 | 2.51 |
| | 0.99 | 2.56 | 69.2 | 8.78 | 46.0 | 70.8 | 2.47 |

### C.1.3 Parameter Setting

Our training and inference configurations for GPT-2 fine-tuning (Table 6) maintain full consistency with the experimental setup in [47]. The model is trained using a linear learning rate schedule over 5 epochs. Optimizer hyperparameters are systematically outlined in Table 7, with two key adaptations based on empirical observations:(1) plain RGD Stabilization: To address the comparatively large learning rate of plain Riemannian Gradient Descent (RGD) and mitigate potential training instability caused by gradient updates, we reduced the weight decay parameters. (2) RAdamW Configuration: The first-order moment parameter $\beta_3$ is fixed at 0.9, while a grid search is performed for the second-order moment parameter $\beta_1 = \beta_2$ within the set $0.96, 0.98, 0.99, 0.999$.

### C.2 Stable Diffusion V1.5

Diffusion models are widely used in image generation tasks, and LoRA has become a common technique for fine-tuning diffusion models. We take the commonly used Stable Diffusion V1.5 model as an example to demonstrate the effectiveness of RAdaGrad in LoRA fine-tuning for object generation. The experiments are conducted on three datasets from Huggingface open-source models [39]: naruto-blip-captions [7], flowers-blip-captions [39], and simpsons-blip-captions, with corresponding prompts "a hellokitty with naruto style", "A woman in long hair in <simpsons>", and "a

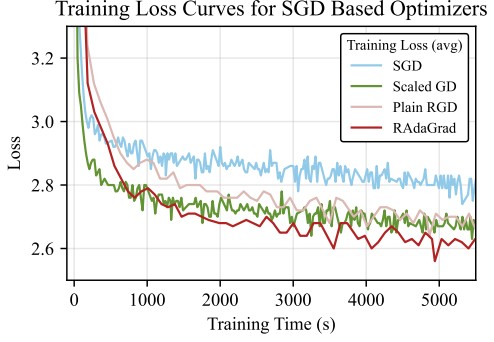

(a) Training loss curve over training time when fine-tuning with rank 8 using SGD-based methods.

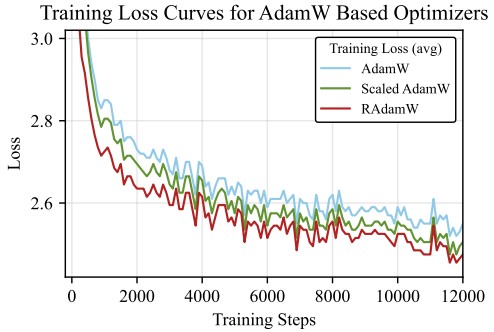

(b) Training loss curve over training step when fine-tuning with rank 64 using AdamW-based methods.

Figure 1: Training loss of GPT-2 small model fine-tuned using different optimizers.

Table 6: Training and Inference Configuration for GPT-2 fine-tuning.

| Parameter | Value | Parameter | Value |
|---|---|---|---|
| **Training** | | **Inference** | |
| Dropout Prob | 0.1 | | |
| Batch Size | 8 | | |
| # Epoch | 5 | Beam Size | 10 |
| Warmup Steps | 500 | Length Penalty | 0.8 |
| LR Scheduler | Linear | No Repeat Ngram Size | 4 |
| Label Smooth | 0.1 | | |
| LoRA $\alpha$ | 32 | | |

Table 7: Core Optimizer Parameters for GPT-2 fine-tuning.

| Methods | Weight Decay | Learning Rate ($\times 10^{-3}$) rank 4 | rank 8 | rank 16 | $\beta_3$ | $\beta_1 = \beta_2$ |
|---|---|---|---|---|---|---|
| SGD | 0.01 | 90 | 90 | 200 | / | / |
| scaled GD | 0.01 | 20 | 30 | 40 | / | / |
| plain RGD | 0.0001 | 80 | 100 | 200 | / | / |
| RAdaGrad | 0.01 | 5 | 8 | 40 | / | 0.98 |
| AdamW | 0.01 | 0.2 | 0.2 | 0.2 | 0.9 | 0.999 |
| scaled AdamW | 0.01 | 0.8 | 2 | 2 | 0.7 | 0.8 |
| RAdamW | 0.01 | 8 | 10 | 8 | 0.9 | 0.98 |

yellow flower". Figure 2 displays the generated results from the four algorithms, with all experiments training LoRA for 4000 epochs.

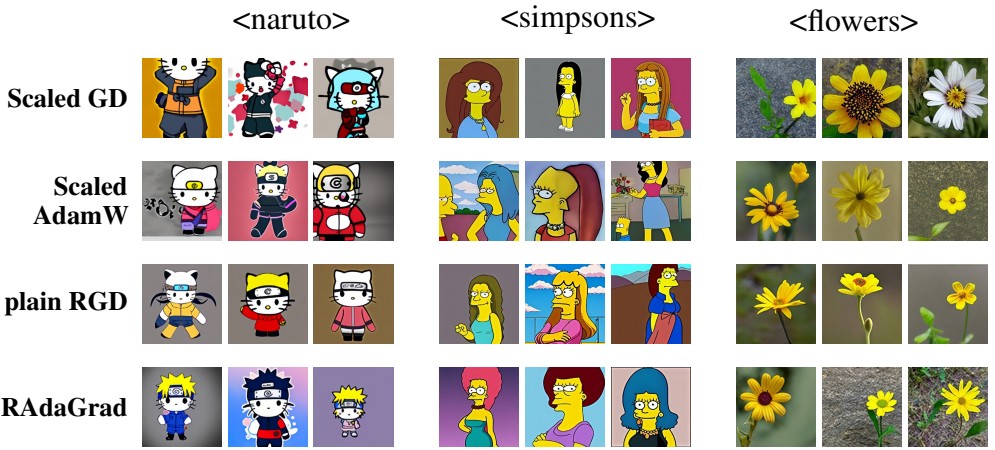

Figure 2: Generation results for LoRA <naruto>, <flowers> and <simpsons> with different optimizers. Default optimizer is AdamW with default learning rate $1e - 4$ for U-Net tuning and $1e - 5$ for text-encoder tuning.

**Simpsons-blip-caption.** The Simpsons-blip-caption dataset contains 786 (image, description) pairs, and Figure 2 shows the final generated results from four algorithms. The image quality produced by all four optimizers is comparable, with each successfully capturing the distinctive style of the Simpsons animation. However, the intermediate results of the generation process highlight the advantages of RAdaGrad. As shown in Figure 3, at the 500th epoch, while the other algorithms had not yet produced clear images, the images generated by RAdaGrad already exhibited the strong stylistic features of the Simpsons, indicating faster convergence and greater efficiency.

### C.3 Mix-Of-Show

We begin by training the model using 14 images of Potter provided by the original project repository, replacing the character names in the image captions with $< V_{Potter} >$. The provided tokens include

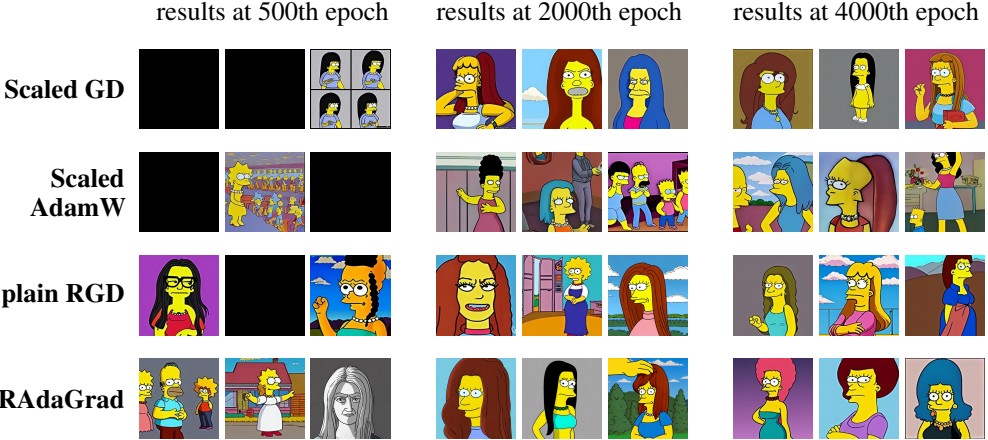

Figure 3: Generation results for LoRA <simpsons> with prompt "A woman with long hair with <simpsons> style." using different optimizers. It shows the changes in objects generated by different algorithms over training time.

Table 8: Hyperparameters for Mix-of-Show fine-tuning

| | plain RGD | | RAdaGrad | | RAdamW | |
|---|---|---|---|---|---|---|
| | 1 | 4 | 1 | 4 | 1 | 4 |
| **Training** | | | | | | |
| Mixed precision | | | fp16 | | | |
| Weight Decay | 0.0 | | 1e-2 | | 1e-4 | |
| Batch Size | | | 2 | | | |
| # Steps | | | 3500 | | | |
| Unet Kv Drop Rate | | | 0 | | | |
| LR Scheduler | | | Linear | | | |
| LR (tuned, $\times 10^{-4}$) | 1e3 | 1e3 | 0.7 | 1.0 | 0.7 | 1.0 |
| AdamW $\beta_3$ | / | | / | | 0.9 | |
| AdamW $\beta_1 = \beta_2$ | / | | 0.999 | | 0.999 | |
| LoRA $\alpha$ | | | 1.0 | | | |
| **Inference** | | | | | | |
| Mixed precision | | | fp16 | | | |
| Num Samples Per Prompt | | | 10 | | | |
| Batch Size | | | 4 | | | |
| Alpha List | | | [0, 0.7, 1.0] | | | |
| Num Inference Steps | | | 50 | | | |
| Guidance Scale | | | 7.5 | | | |

"blurry background", "upper body", "looking at the viewer", "holding a wand", "standing", and others. Figure 4 shows the generation results for the prompt "A $V_{Potter}$ wearing a blue shirt" and Figure 5 shows the generation results for the prompt "A $V_{Potter}$ in front of eiffel tower" The images generated by RAdaGrad and RAdamW more closely resemble Potter. Most images generated by ScaledAdamW resemble Potter, but the eyes in the images are influenced by the word "blue" and appear blue eyes, differing from the real eyes of Potter.

Further, we fine-tune the model using 15 images of Hermione provided by the original repository, replacing the character names in the image captions with $< V_{Hermione} >$. The provided tokens include "blurry background", "upper body", "looking at the viewer", "head tilt", "arms at sides", "brown background", "holding a wand", "standing", and others. Figure 6 shows the results for the prompt "A $< V_{Hermione} >$ wearing a red hat" and Figure 7 shows the generation results for the prompt "A photo of $< V_{Hermione} >$".

The results in Figure 6 indicate that RAdaGrad and RAdamW significantly outperform the other algorithms, especially in capturing facial features. The Hermione wearing a red hat generated by these two algorithms more closely resembles Hermione's real facial characteristics. However, the

facial features in the images generated by Scaled SGD and SGD appear highly unnatural and even fail to resemble Hermione herself. In Figure 7, the images of Hermione generated by RAdaGrad and RAdamW exhibit more distinctive characteristics of Hermione compared to other algorithms. In particular, the images generated by RAdaGrad display richer facial details, such as freckles on Hermione's face, capturing her facial features more precisely. Similarly, RAdamW also captures some details more finely. These all results highlight the powerful image-generation capabilities of RAdaGrad and RAdamW.

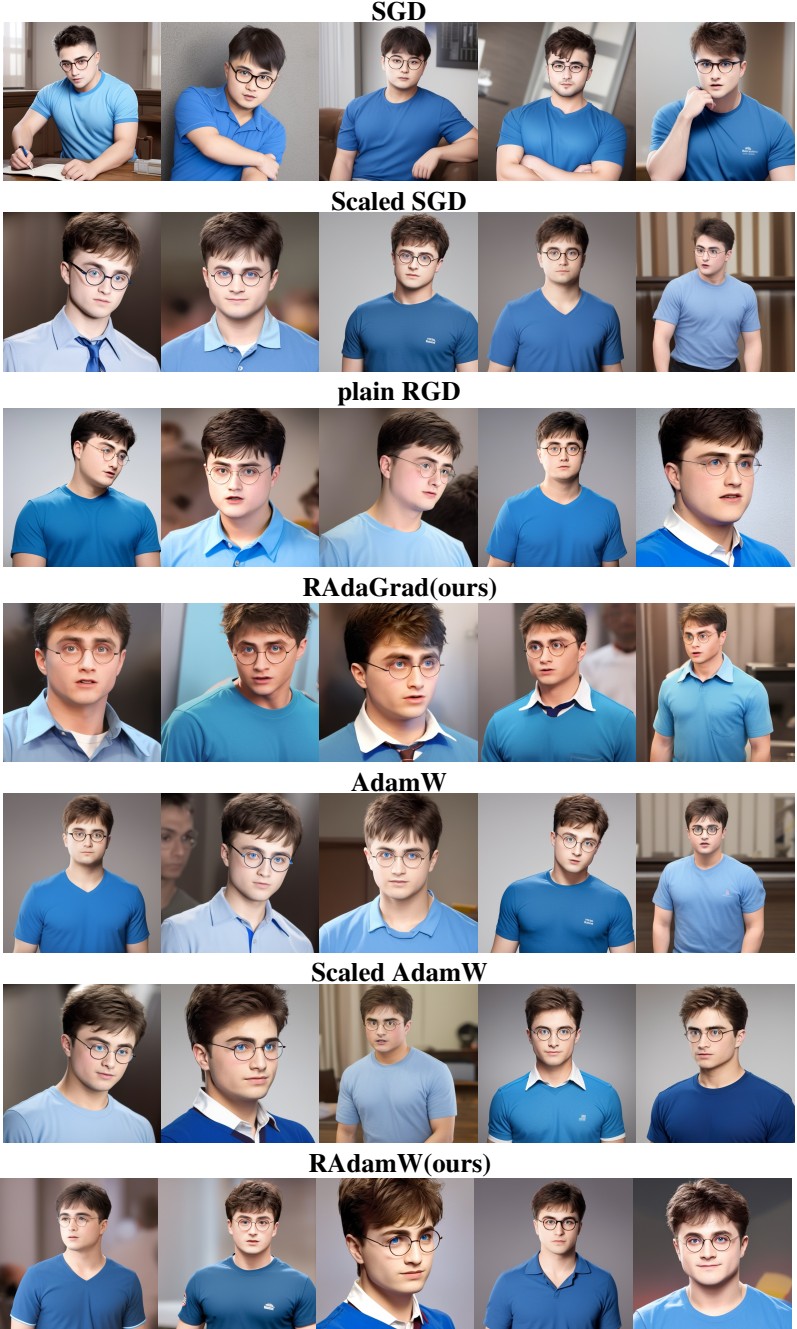

Figure 4: Generation results for LoRA <Potter> with prompt "A $V_{Potter}$ wearing a blue shirt." using different optimizers.

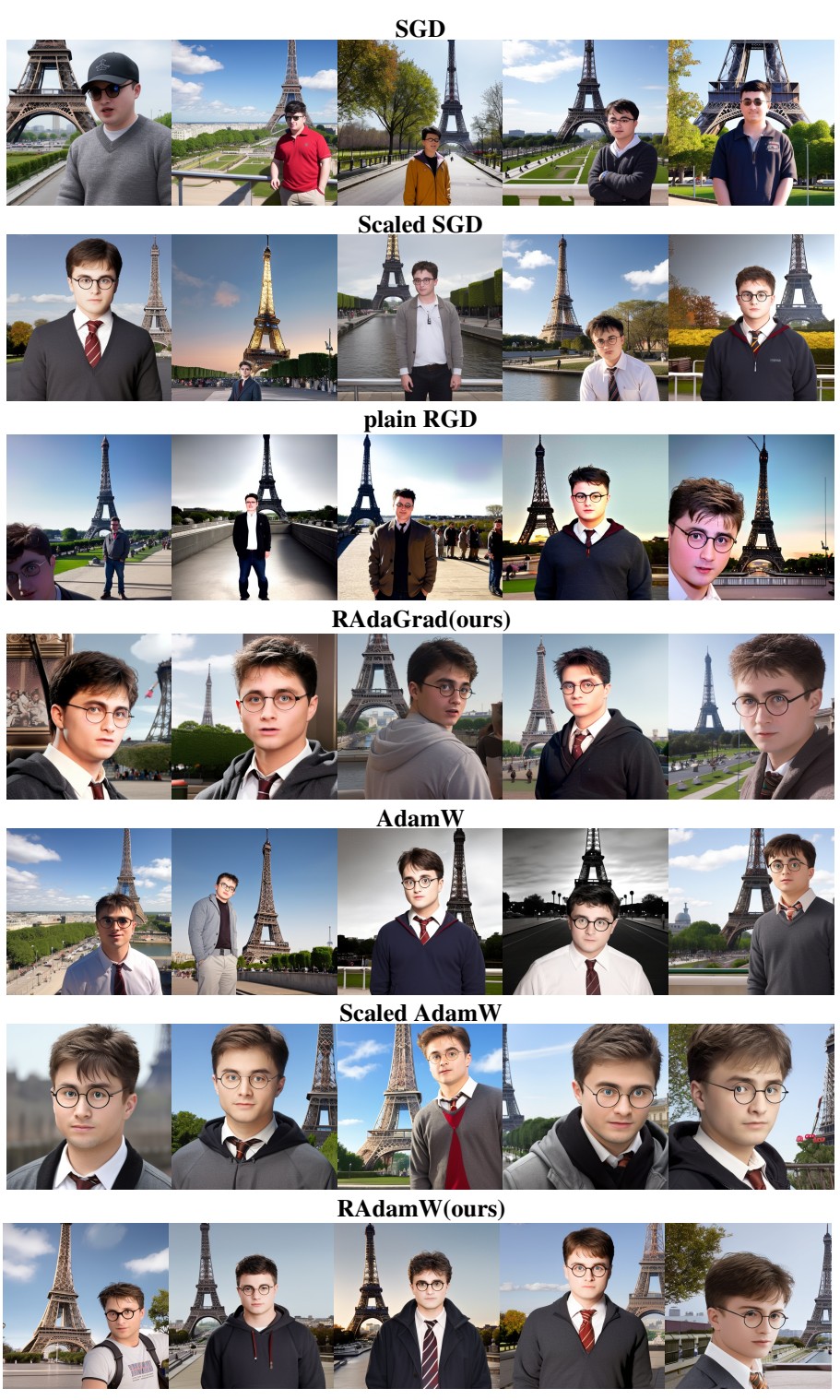

Figure 5: Generation results for LoRA <Potter> with prompt "A $V_{Potter}$ in front of eiffel tower." using different optimizers.

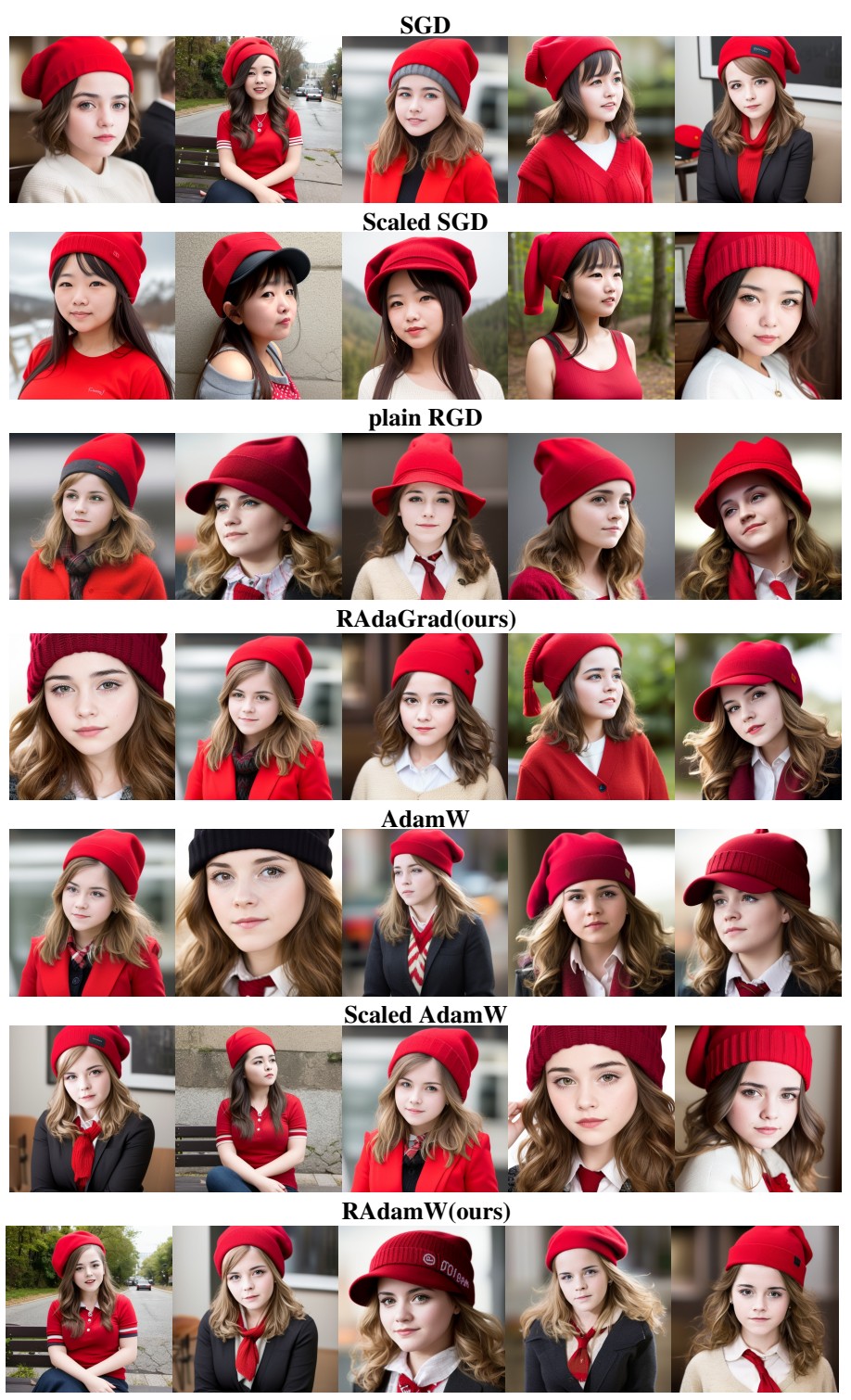

Figure 6: Generation results for LoRA <Hermione> with prompt "A <Hermione> wearing a red hat." using different optimizers.

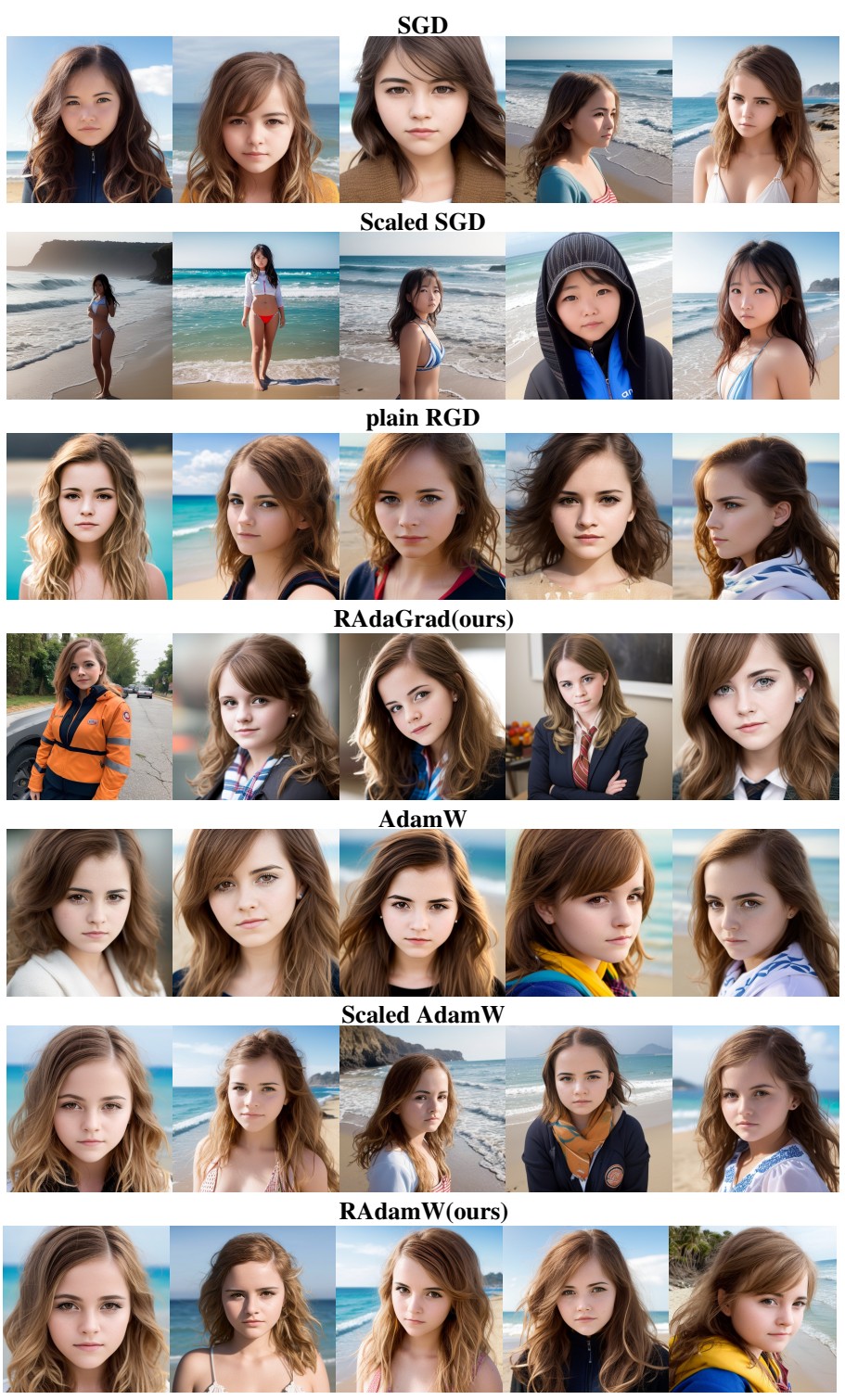

Figure 7: Generation results for LoRA <Hermione> with prompt "A photo of <Hermione>." using different optimizers.

# D Code Implementation Details in Our Paper

The corresponding code for Algorithms 1, 2, and 3 can be found in `Codes/examples/NLG/src/optimizer_custom.py`. The classes corresponding to these algorithms are listed below:

- Algorithm 1 — `class plain RGD(Optimizer)`
- Algorithm 2 — `class RAdaGrad(Optimizer)`
- Algorithm 3 — `class RAdamW(Optimizer)`

---

**Algorithm 1** Pseudocode of plain RGD in Pytorch

---

```python
# Create lora_U, lora_S, lora_Vh in loralib/layer.py
lora_U = nn.parameter(self.weight.zeros(out_features, r))
lora_S = nn.parameter(self.weight.zeros(r))
lora_Vh = nn.parameter(self.weight.zeros(r, in_features))
# Initialize lora_U, lora_S, lora_Vh in loralib/layer.py
nn.init.zeros_(self.lora_U)
nn.init.ones_(self.lora_S)
nn.init.kaiming_uniform_(self.lora_Vh, a=math.sqrt(5))
# Use hook to catch the gradient of W in loralib/layer.py
W.regisiter_hook()
# Group trainable parameters into LoRA pairs in optimizer.py.
for lora_U, lora_S, lora_Vh in pairwise(trainable_parameter):
    param_groups.append({"params": [lora_U, lora_S, lora_Vh], \
        "lr": learning_rate})
# Update rules in optimizer.py
for group in param_groups:
    U, S, Vh = group["params"]
    G_W = W_grad # gradient of W
    # compute the weight update component
    pre_grad = -group['lr'] * G_W
    # compute some matrices for later use
    UU_T_inv = torch.inverse(U.T @ U)
    VV_T_inv = torch.inverse(Vh @ Vh.T)
    # compute matrix K1
    Y1 = UU_T_inv @ U.T @ pre_grad @ (I - Vh.T @ Vh)
    Q1, K1 = torch.linalg.qr(Y1.T)
    # compute matrix K0
    K0 = torch.diag(S) + UU_T_inv @ U.T @ pre_grad @ Vh.T \
        + (U.T - UU_T_inv @ U.T) @ pre_grad @ Vh.T @ VV_T_inv
    # compute matrix K2
    Y2 = (torch.eye(U.shape[0]) - U @ U.T) @ pre_grad @ Vh.T @ VV_T_inv
    Q2, K2 = torch.linalg.qr(Y2)
    # SVD of a matrix of size 2r x 2r
    U_m, S_m, Vh_m = torch.linalg.svd([[K0, K1.T], [K2, 0]])
    # Update parameters
    U = [U, Q2] @ U_m, S = S_m, Vh = Vh_m @ [[Vh], [Q1.T]]
    U = U[:, :self.rank], S = S[ :self.rank], Vh = Vh[:self.rank, :]
```

---

**Algorithm 2** Pseudocode of RAdaGrad in Pytorch

```python
# Create lora_U, lora_S, lora_Vh in loralib/layer.py
lora_U = nn.parameter(self.weight.zeros(out_features, r))
lora_S = nn.parameter(self.weight.zeros(r))
lora_Vh = nn.parameter(self.weight.zeros(r, in_features))
# Initialize lora_U, lora_S, lora_Vh in loralib/layer.py
nn.init.zeros_(self.lora_U)
nn.init.ones_(self.lora_S)
nn.init.kaiming_uniform_(self.lora_Vh, a=math.sqrt(5))
# Use hook to catch the gradient of W in loralib/layer.py
W.regisiter_hook()
# Group trainable parameters into LoRA pairs in optimizer.py.
for lora_U, lora_S, lora_Vh in pairwise(trainable_parameter):
    param_groups.append({"params": [lora_U, lora_S, lora_Vh], \
        "lr": learning_rate})
# Update rules in optimizer.py
for group in param_groups:
    U, S, Vh = group["params"]
    G_W = W_grad # gradient of W
    L_t = beta1*L_{t-1} + (1-beta1)*torch.einsum('ij,ij->i', G_W, G_W)
    R_t = beta2*R_{t-1} + (1-beta2)*torch.einsum('ij,ij->j', G_W, G_W)
    # compute the weight update component
    pre_grad = -group['lr']*[torch.diag(L_t**(-0.25)) @ G_W \
            @ torch.diag(R_t**(-0.25))]
    # compute some matrices for later use
    ULU_inv = torch.inverse(U.T @ torch.diag(L_t**0.25) @ U)
    VRV_inv = torch.inverse(Vh @ torch.diag(R_t**0.25) @ Vh.T)
    ULZ = U.T @ torch.diag(L_t**0.25) @ pre_grad
    ZRV = pre_grad @ torch.diag(R_t**0.25) @ Vh.T
    # compute matrix K1
    Y1 = ULU_inv @ ULZ @ (torch.eye(Vh.shape[0])-Vh.T @ Vh)
    Q1, K1 = torch.linalg.qr(Y1.T)
    # compute matrix K0
    K0 = torch.diag(S) + ULU_inv @ ULZ @ Vh.T + \
        (U.T - ULU_inv @ U.T @ torch.diag(L_t**0.25)) @ ZRV @ VRV_inv
    # compute matrix K2
    Y2 = (torch.eye(U.shape[0]) - U @ U.T) @ ZRV @ VV_T_inv
    Q2, K2 = torch.linalg.qr(Y2)
    # SVD of a matrix of size 2r x 2r
    U_m, S_m, Vh_m = torch.linalg.svd([[K0, K1.T], [K2, 0]])
    # Update parameters
    U = [U, Q2] @ U_m, S = S_m, Vh = Vh_m @ [[Vh], [Q1.T]]
    U = U[:, :self.rank], S = S[ :self.rank], Vh = Vh[:self.rank, :]
```

# E  Supplementary Experiments for DNNs Compression

## E.1  DNNs Compression on MNIST Dataset.

To evaluate the performance of the algorithms, we randomly divide the MNIST dataset [10] into a training dataset with 50,000 samples and a test dataset with 10,000 samples. The image dataset is normalized pixel-wise, without any other data augmentation or regularization techniques applied.

**Five-Layer Fully Connected Network.** We first train a five-layer fully connected neural network on the MNIST dataset. The number of neurons for each layer is $(5120, 5120, 5120, 5120, 10)$. To achieve DNNs compression, we employ a dynamic rank adjustment strategy [25] from DLRT for the weight matrix of each layer. This strategy adaptively truncates the singular values in $S = \text{diag}(\sigma_i)$ based on a parameter $\tau$ and selects the smallest $r \times r$ submatrix that satisfies a specific condition $(\Sigma_{i \geq (r+1)} \sigma_i^2)^{1/2} \leq \tau$. The initial rank for weight matrix of each layer is set to $(500, 500, 500, 500, 10)$. We set the optimal dynamic rank adjustment thresholds $\tau$ for plain RGD, RAdaGrad, and DLRT to be 0.5, 0.5, and 0.13, respectively. Notably, the $\tau$ value for DLRT is smaller because, during experimentation, we observe that larger or smaller $\tau$ values led to a decrease in test accuracy for DLRT. 0.13 is identified as the optimal threshold for maintaining high test accuracy.

**Algorithm 3** Pseudocode of RAdamW in Pytorch

```python
# Create lora_U, lora_S, lora_Vh in loralib/layer.py
lora_U = nn.parameter(self.weight.zeros(out_features, r))
lora_S = nn.parameter(self.weight.zeros(r))
lora_Vh = nn.parameter(self.weight.zeros(r, in_features))
# Initialize lora_U, lora_S, lora_Vh in loralib/layer.py
nn.init.zeros_(self.lora_U)
nn.init.ones_(self.lora_S)
nn.init.kaiming_uniform_(self.lora_Vh, a=math.sqrt(5))
# Use hook to catch the gradient of W in loralib/layer.py
W.regisiter_hook()
# Group trainable parameters into LoRA pairs in optimizer.py.
for lora_U, lora_S, lora_Vh in pairwise(trainable_parameter):
    param_groups.append({"params": [lora_U, lora_S, lora_Vh], \
        "lr": learning_rate})
# Update rules in optimizer.py
for group in param_groups:
    U, S, Vh = group["params"]
    G_W = W_grad # gradient of W
    W_m_t = beta1*W_m_{t-1}+(1-beta1)*G_W # update first moment in step t
    L_t = beta1*L_{t-1} + (1-beta1)*torch.einsum('ij,ij->i', G_W, G_W)
    R_t = beta2*R_{t-1} + (1-beta2)*torch.einsum('ij,ij->j', G_W, G_W)
    # compute the weight update component
    step_size = group['lr'] * sqrt(1-beta2**t) / (1-beta1**t)
    pre_grad = -step_size*[torch.diag(L_t**(-0.25)) @ W_m_t \
                @ torch.diag(R_t**(-0.25))]
    # compute some matrices for later use
    ULU_inv = torch.inverse(U.T @ torch.diag(L_t**0.25) @ U)
    VRV_inv = torch.inverse(Vh @ torch.diag(R_t**0.25) @ Vh.T)
    ULZ = U.T @ torch.diag(L_t**0.25) @ pre_grad
    ZRV = pre_grad @ torch.diag(R_t**0.25) @ Vh.T
    # compute matrix K1
    Y1 = ULU_inv @ ULZ @ (I - Vh.T @ Vh)
    Q1, K1 = torch.linalg.qr(Y1.T)
    # compute matrix K0, K2 similarly
    # ...
    # Update parameters
    # SVD of a matrix of size 2r x 2r
    U_m, S_m, Vh_m = torch.linalg.svd([[K0, K1.T], [K2, 0]])
    U = [U, Q2] @ U_m, S = S_m, Vh = Vh_m @ [[Vh], [Q1.T]]
    U = U[:, :self.rank], S = S[ :self.rank], Vh = Vh[:self.rank, :]
```

Furthermore, we also select the optimal training parameters for the three algorithms. The batch size of plain RGD and RAdaGrad is 128, and the learning rates are 0.07 and 0.06, respectively. The batch size of DLRT is 256 and the learning rate is 0.01.

Table 9: The best test accuracy and the final rank of different methods for a fully-connected network.

| Algorithms | Rank | Test Accuracy |
|------------|------|---------------|
| DLRT | (30,29,28,34,5) | 96.77% |
| plain RGD | (37,27,36,35,5) | 98.38% |
| RAdaGrad | **(16,6,9,24,5)** | **98.65**% |

Under these experimental settings, we train fully connected neural networks using these three algorithms for a total of 75 epochs, and record the test accuracy and loss function, as shown in Figures 8(a) and (b). To more clearly illustrate the effect of DNN compression, we also display the evolution of the rank of weight matrices for each layer during the training process, as shown in Figures 9(a), (c), and (e). The final test accuracy results are shown in Table 9. The experimental results indicate that the RAdaGrad algorithm achieves the highest test accuracy while achieving the maximum DNNs compression rate. The final ranks of the weight matrices obtained by the RAdaGrad algorithm are $(16, 6, 9, 24, 5)$, which are the lowest among the three algorithms, and its test accuracy

reached $98.8\%$, also the highest among the three algorithms. This convincingly demonstrates that compared to the factorized-based algorithm (baseline DLRT), the RAdaGrad algorithm can eliminate the dependence on the condition numbers of two low-rank factors. Furthermore, compared to plain RGD, the RAdaGrad algorithm achieves the same level of test accuracy while providing superior DNN compression.

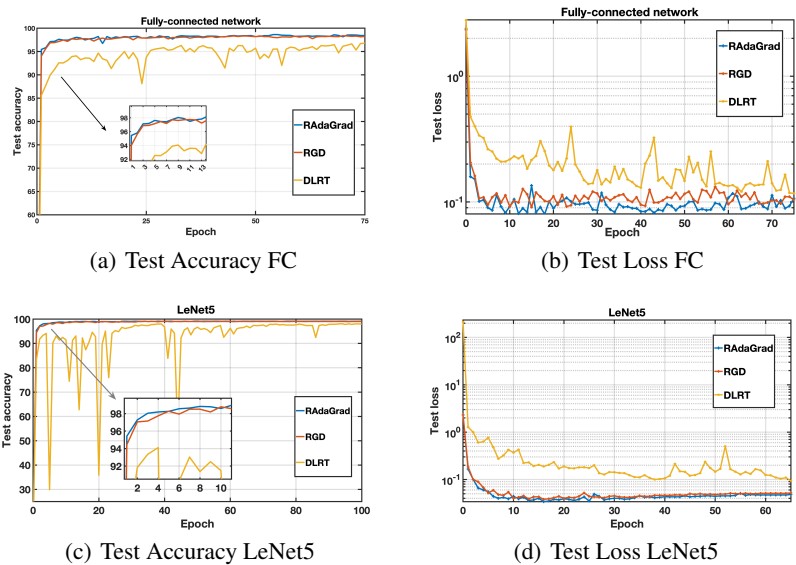

(a) Test Accuracy FC      (b) Test Loss FC

(c) Test Accuracy LeNet5      (d) Test Loss LeNet5

Figure 8: Test accuracy and loss function for the fully connected network and LeNet5 network.

**Convolutional Network – LeNet5.** To further verify the effectiveness of the RAdaGrad algorithm, we apply it to compress the LeNet5 network on the MNIST dataset. For the convolutional layers in LeNet5, we reshaped their convolutional kernels into matrices, with dimensions calculated as: n (number of output channels) $\times$ m (number of input channels $*$ kernel size). Consistent with the fully connected networks compression, we still adopt the dynamic rank adjustment strategy from DLRT. The initial rank for weight matrix of each convolutional layer is set to $r_k = \min(n_k, m_k) / 2$. Through experimentation, we determine the optimal thresholds $\tau$ for the plain RGD, RAdaGrad, and DLRT algorithms to be 0.6, 0.9, and 0.11, respectively. Additionally, we set the optimal training parameters for the three algorithms: both plain RGD and RAdaGrad had a batch size 64 with learning rate 0.11; DLRT had a batch size 128 with a learning rate 0.02.

Table 10: The best test accuracy and the final rank of different methods for LeNet5.

| Algorithms | Rank | Test Accuracy |
|---|---|---|
| DLRT | (7,12,19,5) | 97.231% |
| plain RGD | (10,25,14,5) | 99.2337% |
| RAdaGrad | (10,25,9,5) | **99.2536%** |

We train LeNet5 using these three algorithms for 70 epochs. The test accuracy and test loss are shown in Figures 8(c) and (d), respectively. The evolution of the rank for each layer during the training process, as depicted in Figures 9(b), (d), and (f). The final test accuracy and ranks are listed in Table 10. The results indicate that the RAdaGrad algorithm not only achieved the highest compression rate but also attained the best test accuracy, consistent with the findings from the experiments on fully connected networks. Specifically, the final ranks of the weight matrices achieved by the RAdaGrad algorithm are $(10, 25, 9, 5)$, the lowest among the three algorithms, and its test accuracy reached $99.25\%$, the highest among all algorithms. This further demonstrates that the convergence rate of the RAdaGrad algorithm is independent of the condition numbers of two low-rank factors.

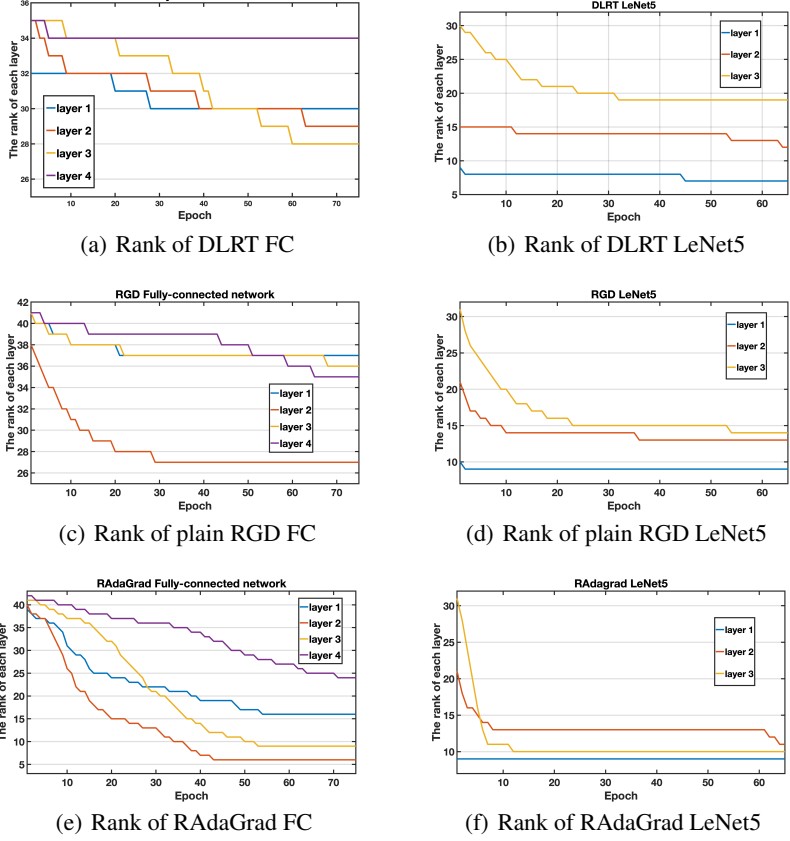

(a) Rank of DLRT FC          (b) Rank of DLRT LeNet5

(c) Rank of plain RGD FC          (d) Rank of plain RGD LeNet5

(e) Rank of RAdaGrad FC          (f) Rank of RAdaGrad LeNet5

Figure 9: The evolution of ranks for each layer in the fully connected network and LeNet5 network.

## E.2 DNNs Compression on CIFAR10 Dataset.

We further test the three algorithms on the CIFAR-10 dataset. The CIFAR-10 dataset consists of 60000 color images, each with a resolution of $32 \times 32$ pixels, divided into 10 classes, with each class containing 6000 images. Among these, 50000 images are used as the training dataset, and 10000 images are used as the test dataset. In this experiment, we train the VGG11 and VGG16 networks with low-rank weight matrices. All algorithms are trained for 400 epochs.

**VGG11 and VGG16.** VGG11 and VGG16 are two widely used deep convolutional neural network models developed by the Visual Geometry Group (VGG) at the University of Oxford. The models are renowned for their excellent performance in image recognition and classification, particularly in the 2014 ImageNet challenge. These models are characterized by the use of small $3 \times 3$ convolutional kernels and multiple convolutional layers. VGG11 includes 11 weight layers, including 8 convolutional layers and 3 fully connected layers, while VGG16 includes 16 weight layers, with 13 convolutional layers and 3 fully connected layers.

Table 11: The best test accuracy and the final rank of different methods for VGG11.

| Algorithms | Rank | Test Accuracy |
|---|---|---|
| DLRT | (10,11,14,14,13,14,12,14,12,21,2) | 66.7623% |
| plain RGD | (2,2,2,2,2,2,2,2,2,2,5) | 90.3877% |
| RAdaGrad | (2,2,2,2,2,2,2,2,2,2,5) | 90.1701 % |

We conduct compression experiments on the VGG11 and VGG16 DNNs using three optimizers: RAdaGrad, plain RGD, and DLRT, with the final results presented in Tables 11 and 12. From the results in these two tables, it can be observed that the compression performance of plain RGD and

Table 12: The best test accuracy and the final rank of different methods for VGG16.

| Algorithms | Rank | Test Accuracy |
|---|---|---|
| DLRT | (7,11,11,12,13,14,15,15,12,15,13,16,15,17,19,5) | 77.2449% |
| plain RGD | (2,2,2,2,2,2,2,2,2,2,2,2,2,2,2,5) | 92.0886% |
| RAdaGrad | (2,2,2,2,2,2,2,2,2,2,2,2,2,2,2,5) | 92.1477% |

RAdaGrad is comparable, and both significantly outperform the DLRT algorithm. Specifically, plain RGD and RAdaGrad achieve extremely low ranks while maintaining high test accuracy. In contrast, the DLRT algorithm results in higher ranks and relatively lower test accuracy. These results strongly support our core idea: optimization algorithms based on RGDframework (such as plain RGD and RAdaGrad) have proven to be an efficient method for DNNs compression by finding low-rank weight matrices on Riemannian manifolds. This kind of method not only significantly reduces the memory usage of DNNs but also largely preserves their performance, achieving excellent results in practical applications. This fully validates our previous statement: optimization algorithms based on the RGD framework not only eliminate the redundancy introduced by matrix factorization methods and completely avoid the negative impact of the Jacobian's condition number, but also effectively reduce the condition number of the Hessian by selecting appropriate metrics on the Riemannian manifold.

