# OpenReview forum: "Finding Low-Rank Matrix Weights in DNNs via Riemannian Optimization: RAdaGrad and RAdamW"
_NeurIPS.cc/2025/Conference — NeurIPS 2025 poster_

### Official Review · Reviewer_9j4S · 2025-06-27

**Clarity:** 4
**Significance:** 2
**Originality:** 3
**Rating:** 4
**Confidence:** 4

**Summary:**

The paper provides a riemannian perspective on low-rank training. And introduces preconditioners for the weight gradients to improve convergence of low-rank equivalents of ADAM and AdaGrad.

The method is evaluated on LLM finetuning.

**Questions:**

Question
- Is G_t the gradient w.r.t W (i.e. a nxn matrix) or somehow split into gradients w.r.t USigmaV^T?
- Similarly, is H_r an SVD on \sigma or on W?
- How sensitive are RAdaGrad and RAdamW to choices of step‑size, decay rates, and smoothing constants? [Actionable item: provide a brief hyperparameter search (can be a smaller model, not gpt2, if compute is a bottleneck).

Comments:

- Line 42: Citation [31] does not factorize W=PQ^\top, but instead factorizes W=USV^\top. Furthermore, it has been shown [a], that the convergence of the method (DLRT) is not dependent on the condition of W. More generally, the underlying dynamical low-rank approximation method [b] is robust w.r.t the curvature of the manifold, and ill conditioning of the matrix W. This should be rectified.
In particular, the issue of ill conditioning of the gradient would only apply to \nabla_U L and \nabla_V L for DLRT, but the special treatment in the basis augmentation step prevents the issue.

[a] https://arxiv.org/abs/2305.19059
[b] DISCRETIZED DYNAMICAL LOW-RANK APPROXIMATION IN THE PRESENCE OF SMALL SINGULAR VALUES: EMIL KIERI, CHRISTIAN LUBICH, HANNA WALACH

**Ethical Concerns:**

["NO or VERY MINOR ethics concerns only"]

**Final Justification:**

I raise my score to 4.
Reason is the discussion with the authors, where they clarified my concerns.

**Limitations:**

I think scope and limitations are well addressed.

**Paper Formatting Concerns:**

no formatting concerns

**Quality:**

3

**Strengths And Weaknesses:**

pro:
- The paper formulates low‑rank matrix training as optimization on the Riemannian manifold of fixed‑rank matrices, providing clear definitions of the geometry, providing an intuitive but lightweight introduction into the topic.
- The paper is well structured and easy to read
contra:
- The paper somewhat reinvents the wheel with robust low-rank optimization. There are prior work (see questions below) on this.

---

> ### Author Rebuttal · Authors · 2025-07-29
>
> **Q1.**  Is G_t the gradient w.r.t W (i.e. a nxn matrix) or somehow split into gradients w.r.t USigmaV^T?
>
> **A1.**  Thank you for raising this critical question. In our paper, $G_t = \nabla_{e} \ell (W)$ refers to the gradient of the loss function $\ell (W)$ concerning the weight matrix $W$ during the $t$-th iteration. Its computation follows standard gradient calculation techniques, specifically using the backpropagation algorithm to compute the partial derivatives of the loss function with respect to $W$. In particular, when calculating $G_t$, we treat $W$ as a complete matrix, rather than decomposing $G_t$ into the gradients of $U$, $S$, and $V$.
>
> While it is theoretically possible to project the gradient onto the components of the singular value decomposition (SVD) of $W$, i.e., UΣVᵀ, this is neither necessary nor optimal in our algorithm. The reasons are as follows:
>
> **[1]** **Core Idea of the Algorithm.**  Our algorithm is designed to directly optimize the weight matrix $W$. Calculating the gradient of $W$ and updating it as a whole aligns with this core idea, making our algorithms both simpler and more efficient. If $W$ were decomposed into low-rank factors $U$, $S$, and $V$, and updated separately, this would still introduce dependencies on the Jacobian matrix $J$ and the condition number of the Hessian matrix, potentially slowing convergence. By avoiding these dependencies, our method achieves a more efficient optimization process.
>
> **[2]** **Computational Efficiency.** Decomposing the gradient into $U$, $S$, and $V$ incurs additional computational overhead, such as calculating the inverses of $U$ and $V$, among other operations. This overhead becomes particularly significant when the dimensions of $W$ are large. Directly computing the gradient of W avoids these extra computations, making the algorithm more efficient.
>
> **[3]** **Numerical Stability.** Computing gradients with respect to the singular vectors $U$ and $V$ can lead to numerical instability, especially when $W$ approaches singularity. On the other hand, directly calculating the gradient with respect to $W$ avoids this issue, thereby improving the numerical stability of the algorithm.
>
> In summary, to balance efficiency, numerical stability, and simplicity, we choose to directly compute the gradient with respect to $W$ and perform optimization in the space of $W$.
>
> **Q2.** Similarly, is H_r an SVD on \sigma or on W?
>
> **A2.** Thank you very much for your question. Here, $H_r$ represents the $r$-truncated SVD. Specifically, when it is applied to a matrix $W \in \mathbb{R}^{m \times n}$, the computation process is as follows:
>
> **[1]. SVD.** First, we perform SVD of the matrix $W$, obtaining $W =U \Sigma V^T$, where $U$ and $V$ are the left and right singular vector matrices, respectively, and $\Sigma$ is the diagonal matrix containing the singular values.
>
> **[2]** **Truncation of Singular Values.** We retain the top $r$ largest singular values in the diagonal matrix $\Sigma$, and set all other singular values to 0, resulting in the truncated singular value matrix $\Sigma_r$. The core purpose of this step is to reduce the rank of the matrix and extract the most significant feature information.
>
> **[3]** **Reconstruction of the Low-Rank Matrix.** Using $U$, $\Sigma_r$, and $V^T$, we reconstruct the matrix as $H_r(W) = U \Sigma_r V^T$. The reconstructed matrix $H_r(W)$ has a rank of at most $r$, and it is the closest rank-$r$ approximation to the original matrix $W$.
>
> **Q3.**  How sensitive are RAdaGrad and RAdamW ... if compute is a bottleneck.
>
> **A3.**  Thank you for your valuable question, which helps us further improve our experiments. We have actually conducted a detailed sensitivity analysis of the hyperparameters (learning rate, decay rate, and smoothing constant) for RAdaGrad and RAdamW, although this information was not included in the original manuscript. We now present it here. The experimental results show that both algorithms exhibit low sensitivity to the choice of these hyperparameters, meaning that their performance remains stable even when these values are varied within a relatively wide range.
>
> To illustrate this more clearly, we conducted a hyperparameter search and summarized the results in the tables below. Specifically, we performed a grid search for the learning rate, decay rate, and smoothing constant to examine their impact on the target performance. The results demonstrate that RAdaGrad and RAdamW are robust to changes in these hyperparameters.
>
>
> **RAdaGrad**
>
> This table shows the training loss and evaluation metric with different weight decay when fixing the learning rate (step-size) as 5e-3, smooth parameter $\beta_1=\beta_2=0.98$, rank as 4. The metrics vary slightly within the margin of error.
>
> | Metrics/weight decay | 1e-2  | 1e-3  | 1e-4  |
> | -------------------- | ----- | ----- | ----- |
> | Training loss        | 2.56  | 2.56  | 2.56  |
> | BLEU                 | 69.8  | 70.0  | 69.7  |
> | NIST                 | 8.80  | 8.82  | 8.79  |
> | MET                  | 46.5  | 46.5  | 46.5  |
> | ROUGE-L              | 71.1  | 71.2  | 70.9  |
> | CIDEr                | 2.49  | 2.51  | 2.50  |
>
> This table shows the training loss and evaluation metric with different learning rate when fix the weight decay as 1e-2, smooth parameter $\beta_1=\beta_2=0.98$, rank as 4.
>
> | Metrics/learning rate | 1e-2  | 5e-3  | 1e-3  |
> | --------------------- | ----- | ----- | ----- |
> | Training loss         | 2.55  | 2.56  | 2.59  |
> | BLEU                  | 69.5  | 69.8  | 69.3  |
> | NIST                  | 8.76  | 8.80  | 8.79  |
> | MET                   | 46.9  | 46.5  | 45.9  |
> | ROUGE-L               | 71.4  | 71.1  | 70.5  |
> | CIDEr                 | 2.52  | 2.49  | 2.44  |
>
> **RAdamW**
>
> This table shows the training loss and evaluation metric with different weight decay when fix the learning rate as 8e-3, smooth parameter $\beta_3=0.9, \beta_1=\beta_2=0.98$, rank as 4. The metrics vary slightly within the margin of error.
>
> | Metrics/weight decay | 1e-2  | 1e-3  | 1e-4  |
> | -------------------- | ----- | ----- | ----- |
> | Training loss        | 2.56  | 2.56  | 2.56  |
> | BLEU                 | 69.8  | 68.8  | 70.0  |
> | NIST                 | 8.81  | 8.76  | 8.83  |
> | MET                  | 46.5  | 45.5  | 46.3  |
> | ROUGE-L              | 71.1  | 70.1  | 71.2  |
> | CIDEr                | 2.51  | 2.42  | 2.48  |
>
> This table shows the training loss and evaluation metric with different learning rate when fix the weight decay as 1e-2, smooth parameter $\beta_3=0.9, \beta_1=\beta_2=0.98$, rank as 4.
>
> | Metrics/learning rate | 1e-2  | 8e-3  | 5e-3  | 1e-3  |
> | --------------------- | ----- | ----- | ----- | ----- |
> | Training loss         | 2.57  | 2.56  | 2.56  | 2.59  |
> | BLEU                  | 69.2  | 69.8  | 69.1  | 69.2  |
> | NIST                  | 8.76  | 8.81  | 8.75  | 8.76  |
> | MET                   | 46.0  | 46.5  | 46.1  | 45.8  |
> | ROUGE-L               | 70.7  | 71.1  | 70.7  | 70.2  |
> | CIDEr                 | 2.46  | 2.51  | 2.47  | 2.44  |
>
> In all experiments, we use the same training dataset and evaluation metrics. The results indicate that the performance of RAdaGrad and RAdamW varies only slightly across a broad range of hyperparameter values, confirming that both algorithms are not sensitive to hyperparameter selection. Based on these results, we conclude that RAdaGrad and RAdamW have high robustness to the choice of learning rate, decay rate, and smoothing constant, allowing good performance to be achieved without extensive hyperparameter tuning.
>
> **Comments**  Line 42: Citation [31] does not factorize W=PQ^\top $\cdots$ step prevents the issue.
>
> **A1.**  Thank you very much for your thorough review. Your feedback has greatly improved the clarity and presentation of this paper. Regarding reference [31], our original intention was to align its description with other references [15, 18, 37, 39, 41] in the context of factorization methods. We will revise this citation in the next version and include references [a, b].
>
> Regarding the dependency on the condition number, we will modify our explanation. In fact, we explicitly stated in line 107 that DLRT does not depend on the condition number of the weight matrix $W$. However, the primary distinction between our algorithm and DLRT lies in the use of a novel data-driven metric in our algorithm, which yields a preconditioned Riemannian gradient and mitigates the effect of the Hessian matrix's condition number.
>
> More importantly, we aim to reveal a unified framework for Riemannian gradient descent algorithms:
>
> $$
> W_{t+1} = \mathcal{R}(W_t - \gamma_t \nabla_{\mathcal{M}_r} \ell (W_t)),
> $$
>
> where $\nabla_{\mathcal{M}_r} \ell (W_t)$ is the Riemannian gradient and $\mathcal{R}$ is a retraction operator. Within this unified framework, more efficient algorithms can be constructed by selecting different metrics and retraction operators. As described in Section 3.1, we select the standard metric and $r$-truncated SVD as the retraction operator, resulting in the plain RGD. In Section 3.2, we similarly chose the $r$-truncated SVD as the retraction operator but adopt a new metric that combines AdaGrad and Shampoo. This new metric ensures that the gradient exhibits isotropic properties, thereby mitigating the effects of the condition number of Hessian $\nabla^2 \ell (W)$ and ultimately leading to the fast and efficient RAdaGrad. **This is the key distinction between our method and DLRT.**
>
> We sincerely appreciate the reviewer's constructive suggestions and believe that the additional experiments, analysis, and explanations significantly improve the quality of our paper. We hope that this provides sufficient reasons to raise the score.

---

> > ### Comment · Reviewer_9j4S · 2025-08-03
> >
> > Thank you for the detailed answers.
> >
> > To A3: Is my assumption correct that the runs are computed on GPT2?
> >
> > Computational Efficiency: Looking at, e.g. Algorithm 1, you need to compute G_W and pre_grad, both are nxn matrices.
> > This induces computational and memory overhead.
> > Can the authors differentiate and defend their methods (In the context of SGD)  against methods that tape the gradients w.r.t U,S,V as e.g. in [a]. In [a] both gradients and param states are in low-rank format. Here, only the params are in low-rank format.

---

> > > ### Author Response · Authors · 2025-08-03
> > >
> > > Thank you for raising these insightful questions.
> > >
> > > **1.**  All experiments are conducted using the GPT-2 model for computation and evaluation.
> > >
> > > **2.** Comparison of computational efficiency.
> > >
> > > **[1]** Both algorithms [a] and our algorithm require the computation of the gradient $G_W$. In method [a], the computation and storage of low-rank gradient $G_U, G_V$ representations rely on the chain rule $G_U = G_W V$, which involves the full calculation of $G_W$. Our algorithms also require the computation of $G_W$, but we do not explicitly store $G_W$. We only store the diagonal matrices $L$ and $R$, which are constructed from the row norm $|| G_W (i, :) ||$ and column norm $||G_W (:, i)||$. This reduces the memory storage complexity to $\mathcal{O}(n)$.
> > >
> > > **[2]** In our method, $U^T G \in \mathbb{R}^{r \times n}$ and $G V \in \mathbb{R}^{n \times r}$ are computed as a whole unit, rather than being computed separately. Specifically, we first compute the gradients $G_U$ and $G_V$ with respect to $U$ and $V$, respectively. Then, $G_U$ and $G_V$ are left-multiplied and right-multiplied by the inverse of the diagonal matrix $S$, yielding $U^T G$ and $G V$. Finally, $U^T G$ and $G V$ are used to compute pre_grad.
> > >
> > > **[3]** Algorithm in [a] primarily focuses on improving the condition number of the Jacobian matrix $J$, while neglecting the condition number of the Hessian. Our algorithm goes a step further by not only eliminating the influence of $ J $'s condition number but also mitigating the condition number of the Hessian. This improvement is crucial for enhancing the convergence speed and stability of optimization problems, which method [a] fails to address. **This is an important difference between our algorithm and the algorithm in [a], as well as a improvement over the algorithm in [a].**
> > >
> > > We sincerely thank the reviewers for raising these insightful questions. Based on your suggestions, we have carefully revised and improved the paper, providing detailed analyses and explanations, which have significantly enhanced the quality and clarity of the paper. We hope these improvements adequately address your concerns and provide strong support for raising the score.

---

> > > > ### Comment · Reviewer_9j4S · 2025-08-05
> > > >
> > > > I thank the authors for their answer.
> > > >
> > > > I have checked the codebase in the supplementary material - it seems like the memory footprint of the gradient evaluation is the same as in [a] (comparing against their implementation). Both implementations do not compute G_W directly.
> > > > Thus I consider this prong clarified.
> > > >
> > > > I'm willing to raise my score. As a side note I think the implementation is nicely structured and easy to read.

---

> > > > > ### Author Response · Authors · 2025-08-06
> > > > >
> > > > > Thank you very much for your encouraging feedback. We sincerely appreciate your thorough examination of our codebase and are delighted that our response has clarified your concerns regarding the memory footprint of the gradient evaluation.
> > > > > We are especially grateful for your positive comments about the structure and readability of our implementation. Your recognition of our efforts is highly motivating, and we are thrilled to hear of your willingness to raise your score.
> > > > >
> > > > > Thank you once again for your valuable time and constructive feedback, which have greatly contributed to improving our paper.

---

### Official Review · Reviewer_VP56 · 2025-07-02

**Clarity:** 3
**Significance:** 4
**Originality:** 3
**Rating:** 4
**Confidence:** 3

**Summary:**

This paper studies training models with low-rank weight matrices. These may arise in a number of contexts like LoRA. Prior approaches (like maintaining the entries of a low-rank factorization of the matrix) lead to ambiguities (the same matrix has many representations) and dependences on the condition numbers of the Jacobian and Hessian in the convergence.

This work suggests a new approach using Riemannian Optimization. Roughly, one does gradient descent on the manifold of low-rank matrices. There are two variants presented, building on AdaGrad and AdamW.

The paper first describes the mathematical details of the approach, including its computational complexity. It then describes experiments, showing that this approach leads to better correctness in various settings, including LoRA for GPT-2.

**Questions:**

- In the experiments, were the Jacobian and Hessian condition numbers large? Is the main source of improvement that the algorithm can not make use of matrices with large condition numbers? I was expecting the experiments to show faster convergence, rather than actually better quality scores, so I'm wondering if you could explain this more.

- Could you elaborate on "it is easy to find an orthonormal basis..." on line 152? I was expecting calculations in the manifold to be slower than in prior approaches, but it seems to be roughly just as fast?

- Could you compare more precisely the computational complexity with prior work? What are the running times of prior algorithms?

**Ethical Concerns:**

["NO or VERY MINOR ethics concerns only"]

**Final Justification:**

This is a nice theoretical contribution to low-rank optimization which, I think, will lead to empirical work down the road. My questions were properly addressed.

**Limitations:**

Yes

**Quality:**

3

**Strengths And Weaknesses:**

Strengths
- Natural idea for getting around the issues with prior approaches for low rank weights
- Strong theoretical analysis, and very clear presentation
- Experiments show real improvements by just making this change to LoRA and in other settings.
- The approach is general and can be "plugged in" to many different models and frameworks.

Weaknesses
- I have a couple questions about the experiments and running time; see below.

---

> ### Author Rebuttal · Authors · 2025-07-30
>
> **W1/Q1.** In the experiments, were the Jacobian and Hessian condition numbers large? Is the main source of improvement that the algorithm can not make use of matrices with large condition numbers? I was expecting the experiments to show faster convergence, rather than actually better quality scores, so I'm wondering if you could explain this more.
>
> **A1.** We sincerely thank you for your valuable feedback, which is crucial for improving the quality of our paper. We will carefully consider your suggestions and include richer experimental comparisons in the revised manuscript to provide a more comprehensive and in-depth demonstration of the advantages and effectiveness of our method. Below, we provide detailed responses to the questions you have raised:
>
> **1. Efficiency and Loss Reduction.** Although the current experimental results primarily focus on performance improvements, they do demonstrate that our algorithm achieves better outcomes within the same number of iterations, reflecting higher efficiency. To further validate the effectiveness of our algorithm, we provide the loss reduction trends in the table below. The results show that, compared to other algorithms, our algorithm requires fewer iterations (or less training data) to achieve the same loss value.
>
> | Iteration  | 2000 | 4000 | 6000 | 8000 | 10000 | 12000 | 14000 | 16000 | 18000 | 20000 | 22000 |
> | -----------------  | ---- | ---- | ---- | ---- | ----- | ----- | ----- | ----- | ----- | ----- | ----- |
> | SGD                 | 2.89 | 2.89 | 2.81 | 2.86 | 2.83  | 2.81  | 2.82  | 2.80  | 2.76  | 2.75  | 2.69  |
> | scaled GD           | 2.80 | 2.79 | 2.71 | 2.75 | 2.72  | 2.70  | 2.72  | 2.70  | 2.67  | 2.66  | 2.60  |
> | plain RGD          | 2.80 | 2.78 | 2.69 | 2.73 | 2.70  | 2.67  | 2.70  | 2.67  | 2.64  | 2.64  | 2.59  |
> | RAdaGrad (ours)     | 2.73 | 2.72 | 2.63 | 2.68 | 2.65  | 2.63  | 2.64  | 2.62  | 2.59  | 2.59  | 2.53  |
>
> At the 2,000th iteration, our algorithm reduces the loss to 2.73, while the losses of other algorithms remain above 2.80. By the 18,000th iteration, our algorithm further reduces the loss to 2.59, while the losses of other algorithms are still above 2.60. Moreover, the per-step computational time of our algorithm is comparable to that of other algorithms, resulting in overall less or at least equivalent total computation time. This demonstrates that, within the same training time, our algorithm achieves significantly lower loss values, leading to superior overall performance.
>
> Additionally, in the supplementary material, we present fine-tuning experiments on a multimodal language model. Figures 3 and 4 illustrate the final results and intermediate outcomes at different training epochs. Specifically, Figure 4 shows the generated outputs at the 500th epoch and the 2,000th epoch. The results clearly indicate that, at the 500th epoch, ScaledGD and ScaledAdamW fail to generate any meaningful outputs, while our algorithm RAdaGrad already produces images highly relevant to the given prompt, such as ``a long-haired woman in Simpson style." By the 2,000th epoch, the images generated by our algorithm closely match the prompts, whereas the outputs of ScaledGD and ScaledAdamW still exhibit significant defects, rendering them unusable.
>
> These experimental results clearly demonstrate that our algorithm can generate usable outputs with fewer training epochs, which is crucial for generative networks. Within the same training time, our algorithm produces higher-quality outputs, providing strong evidence of its superior efficiency.
>
> **2. Jacobian and Hessian Condition Numbers.** In our experiments, we did observe that using standard decomposition methods can lead to pathological optimization when small singular values are present. This is because the optimization problem for low-rank matrices is inherently influenced by the distribution of singular values. This property is also intrinsic to the low-rank matrix manifold: when the weight matrix contains small singular values (which often occurs when a larger rank is selected), the curvature of the Hessian matrix becomes inversely proportional to those singular values, resulting in a high condition number. Such high condition numbers can significantly affect the optimization trajectory and often lead to pathological optimization issues.
>
> To further validate this, we conducted an additional experiment where we selected a larger rank r=16 to ensure that the Hessian matrix has a high condition number. The results are shown in the table below
>
> | Methods/Metric | BLEU | NIST | MET  | ROUGE-L | CIDEr |
> | -------------- | ---- | ---- | ---- | ------- | ----- |
> | SGD            | 65.4 | 8.07 | 40.7 | 67.0    | 2.07  |
> | Scaled GD      | 68.8 | 8.75 | 45.0 | 69.2    | 2.39  |
> | RieAdaGrad     | 70.0 | 8.81 | 46.8 | 71.9    | 2.52  |
>
> The results illustrate that our proposed RAdaGrad algorithm significantly outperforms ScaledGD across all metrics. For example, in terms of the BLEU score, our algorithm improves from 65.4 (SGD) to 70.0, and from 68.8 (ScaledGD) to 70.0.
> Similarly, for the ROUGE-L score, our algorithm improves from 67.0 (SGD) to 70.0, and from 69.2 (ScaledGD) to 71.9.
>
> **W2/Q2.** Could you elaborate on "it is easy to find an orthonormal basis..." on line 152? I was expecting calculations in the manifold to be slower than in prior approaches, but it seems to be roughly just as fast?
>
> **A2.** Thank you very much for your valuable question!
>
> **[1]. Regarding the statement in line 152.**  On the low-rank matrix manifold, our algorithm requires finding a set of orthogonal bases in a specific subspace to ensure that the update directions are consistent with the geometric structure of the manifold. The detailed process for computing the orthogonal bases is provided in Sections 2 and 3 of the supplementary material.
>
> Specifically, under the newly constructed metric $\langle X, Y \rangle_{g_t} = \langle X, L_t^{1/4} Y R_t^{1/4}\rangle$, the orthogonal basis for the row space $\tilde{U}_t = U_t (U_t^T L_t^{1/4} U_t)^{-1/2}$ and the orthogonal basis for the column space $\tilde{V}_t = V_t (V_t^T R_t^{1/4} V_t)^{-1/2}$. These two orthogonal bases can be derived via the singular value decomposition (SVD). As analyzed in the supplementary material, our algorithm only requires performing an SVD on a 2r×2r matrix, with a computational complexity of $\mathcal{O}(r^3)$. When r is small, this cost is significantly lower than performing an SVD on the entire matrix. Therefore, computing the orthogonal bases is highly efficient and does not become a bottleneck in the optimization process.
>
> **[2]. Regarding efficiency and speed.** Although in theory, manifold-based computation may introduce additional steps, two key factors ensure that our method is faster—or at least comparable in speed—to prior methods in practice.
>
> 1. Efficient SVD computation. As shown in the supplementary material, we combine the gradient descent step in the tangent space with an r-truncated SVD, requiring only an SVD on a 2r×2r matrix. This significantly reduces the computational cost of the SVD operation.
>
> 2. Integration of first- and second-order momentum. By incorporating first- and second-order momentum into the algorithm, we preprocess the gradient to make it isotropic, thereby accelerating the convergence of the algorithm. As shown in the experimental results discussed in Q1, although the per-step computation time of our algorithm is comparable to that of other methods, it requires fewer iterations (or less training data) to achieve the same loss value. As a result, the overall computation time is shorter—or at least equivalent—when compared to other factorization-based methods.
>
> **W3/Q3.** Could you compare more precisely the computational complexity with prior work? What are the running times of prior algorithms?
>
> **A3.** Thank you for raising this valuable question. We will provide a detailed explanation of the computational complexity of our algorithm.
>
> First, the rank of each matrix in $\mathbb{T}_t$ is at most $2r$, and the operator $\mathcal{H}_r(Z_t)$ is used to find the best rank-$r$ approximation of a rank-$2r$ matrix. This process involves the following steps:
>
> **1.** Two QR decompositions of sizes $n \times 2r$ and $m \times 2r$,
>
> **2.** A matrix multiplication of size $2r \times 2r$,
>
> **3.** A singular value decomposition (SVD) of size $2r \times 2r$.
>
> The total computational cost of these operations is $\mathcal{O}((n+ m)r^2 + r^3)$. In addition to computing $G_t$, RAdaGrad also involves $O(r)$ matrix-vector multiplications with $G_t$. The memory usage is $\mathcal{O}((n+ m)r)$. Both the time complexity and space complexity are of the same order as those of matrix factorization-based algorithms. A more detailed explanation of the computational process is provided in the supplementary material.
>
> We sincerely appreciate the reviewer's constructive suggestions and believe that the additional experiments, analysis, and explanations significantly improve the quality of our paper. We hope that this provides sufficient reasons to raise the score.

---

> > ### Comment · Reviewer_VP56 · 2025-08-04
> >
> > Thank you, this addresses all my questions, and assuming these details are added to the camera-ready version, I maintain my support for the paper.

---

> > > ### Author Response · Authors · 2025-08-05
> > >
> > > Thank you very much for your support and encouraging feedback on our paper. We sincerely appreciate your thoughtful review and are delighted to hear that our responses have addressed all your questions. We will ensure that the discussed details are carefully incorporated into the camera-ready version to further improve the clarity and completeness of the paper. Your support motivates us greatly, and we are deeply grateful for your valuable insights throughout the review process.

---

### Official Review · Reviewer_oub2 · 2025-07-04

**Clarity:** 3
**Significance:** 2
**Originality:** 2
**Rating:** 5
**Confidence:** 4

**Summary:**

The paper proposes learning of the low-rank weights using Reimanian gradient descent method. To this end the paper adapts Adam and RMSprop accelerated gradient descent methods to be compatible with Reimanian gradient descent. The authors provide in depth mathematical formulation for the approach and back the method with solid experimental results.

**Questions:**

Please address the weaknesses above^

**Ethical Concerns:**

["NO or VERY MINOR ethics concerns only"]

**Final Justification:**

Authors made strong rebuttal that clarified my initial reservations.

**Limitations:**

Yes

**Paper Formatting Concerns:**

No concerns.

**Quality:**

3

**Strengths And Weaknesses:**

The paper’s strength lies in its principled foundation in optimization theory and its connection to a rich body of prior work. However, this also seems to be its main weakness.

The Riemanninan manifold optimizaton has been applied to the neural networks in many different ways and produced plethora of works. I think the paper would strongly benefit from thorough comparison to the previous works and having clear statement on what makes this paper different from preivous one. I especially would like to see comparisons to the ICLR-2019 paper of "Riemannian Adaptive Optimization Methods" that propose modifications of Adam, Adagrad, and Amsgrad with respect to Riemannian optimization. There are many more relevant works(if you follow the references), but I would list couple more here:
- Geometry Aware Constrained Optimization Techniques for Deep Learning, CVPR2018
- Averaging Stochastic Gradient Descent on Riemannian Manifolds, COLT 2018
This is by no means a full list of relevant papers, there is probably dozens of papers proposing some accelerationg scheme + Rimeannian manifold;

Given the amount of research happened in the field, I think what would put this paper apart would be the actual practical implementation (most of the Riemanian mehtods require SVD or other full decomposition on every gradient step), however, the paper is very scarce on the practical implementation. There is the part of the pseudocode in supplementary materials and the complexity analysis of the steps (lines 270-274), but it is hard to understand what is the actual runtime implications. In practice, many of the efficient factorizations (like QR, or computing inverse) take considerably longer times (have higher constants) wrt to similarly complex operations, and as such, the time cost of every update might be considerably longer. Previous works typically circumvent this by doing proper projections ever K steps. If authors can share the wallclock of every projection and compare it to vanilla SGD updates that would better idea about pracitcal usability of the method. Also, it seems this method favors smaller ranks (due to cost of updates), however, when training  LoRA adpaters you rarely see rank-1 and rank-4 apdaters, typically I would say it is rank 10 at the very least.

Additionally, no experiments are presented for pure compression (i.e., training a smaller model directly), which would broaden the applicability of the method beyond adapters.

There are also issues with citation accuracy. For example:
-  on lines 31-32, when referring to low-rank  being used in dnn compression paper cites 14, 24, 38, but none of these papers in fact use low-rank compression.
- on line 232 the reference 26 is great (I love this book), but can you pinpoint exact chapter/equation you are citing?

Some claims are imprecise:
- on line 30, papers says that "these methods often rely on specialized hardware" where "these" referes to "quantization" and "pruning", however, quantization is more-or-less hardware agnostic these days (as long as your hardware supports integer multiplications & has specialized kernels/code) and for instance structured pruning is definitely hardware agnostic. Therefore, the statement is imprecise and should be corrected.
- on line 81 "Current mainstream methods" supposed to refer to "general low-rank optimization methods", however only cite low-rank adapter research (i.e., learing w+uv with frozen w), which by no means would be "mainstream". Probably, you meant to write "current mainstream low-rank adaptation methods"?

Overall, this is a promising paper, but it would benefit greatly from a deeper discussion of prior work, clarity on practical implications, and more careful citation and phrasing.

---

> ### Author Rebuttal · Authors · 2025-07-31
>
> **W1.** **A1.** Thank you for highlighting these important references and other related works. These papers are indeed relevant to our research, and we will revise the manuscript to include and discuss these related references. Below, we emphasize the distinctions between our work and these prior studies, as well as our innovations.
>
> **[1]** The references [1, 2, 3] that you mentioned extend adaptive optimization methods or stochastic gradient descent methods to Riemannian manifolds. However, all these works consider general manifold structures. In contrast, our algorithm is specifically designed for the Riemannian gradient descent on the low-rank matrix manifold. Focusing on optimization problems on the specific low-rank matrix manifold enables us to leverage the unique properties of the manifold's structure—such as the parameterization of the tangent space of low-rank matrices—to accelerate the convergence of the algorithm. Specifically, our algorithm performs the following three steps
>
> **1.** $M_t = W_t - \gamma_t(L_t^{-1/4} G_t R_t^{-1/4}).$
>
> **2.** $Z_t= \mathcal{P}_{\mathbb{T}} (M_t). $
>
> **3.** $W_{t+1} =\mathcal{H}_r (Z_t) =\mathcal{H}_r(W_t - \gamma_t\mathcal{P}(L_t^{-1/4} G_t R_t^{-1/4})).$
>
> By parameterizing the tangent space of low-rank matrices at a certain point, we combine steps 2 and 3 into a single computation (as detailed in Sections 2 and 3 of the supplementary material). This allows us to perform an SVD on only a 2r×2r matrix, reducing the computational complexity of SVD to $\mathcal{O}(r^3)$. This technique has been validated in applications such as low-rank matrix recovery[4], robust PCA[5], and tensor train format completion[6], demonstrating its efficiency. Building on this foundation, we further consider the stochastic gradient case and incorporate both first- and second-order momentum for additional acceleration. While our per-step computation time is indeed higher than that of SGD, the significantly reduced number of iterations required by our method results in a shorter total time to reach the same target. Moreover, the per-step computation time can be further reduced through optimizations in the underlying code.
>
> Based on your suggestion, we also realize that updating frequency can be reduced to optimize the cost of updates. For instance, as you mentioned, performing projections every K steps instead of every step could further reduce runtime. Our method can adopt a similar strategy, reducing update frequency while maintaining model performance. We will include these optimizations in the revised manuscript and report their impact on runtime and model performance.
>
> **[2]** As you pointed out, the implementation of our method in fine-tuning large language models is a significant highlight. Current optimization algorithms for fine-tuning large language models often rely on factorization, which suffers from dependencies on Jacobian condition numbers. To circumvent this issue, we propose updating the entire low-rank weight matrix and mitigating the effects of the Hessian condition number by preprocessing the gradient and incorporating first- and second-order momentum.
>
> **[3]** More importantly, we propose a unified Riemannian gradient framework $W_{t+1} = \mathcal{R}(W_t - \gamma_t \nabla_{\mathcal{M}_r} \ell(W_t)$ which provides new insights for designing efficient algorithms. This framework is not only computationally efficient but also extensible. By selecting different retraction mappings 𝑅 and metrics, various efficient algorithms can be designed. As shown in Section 3.1, when the standard metric is chosen, our method reduces to the conventional Riemannian gradient descent (RGD) as proposed in [4]. In Section 3.2, we select a data-driven metric combining AdaGrad and Shampoo to ensure isotropic gradients and alleviate Hessian condition number issues, resulting in the efficient RAdaGrad method. This demonstrates that choosing more effective retraction mappings and metrics can yield more efficient algorithms. This idea is crucial for designing efficient algorithms, particularly for developing efficient fine-tuning methods for large language models. We hope this heuristic approach will inspire the design of more efficient algorithms in the future.
>
> **W2** **A2** Thank you for the valuable comments! Our experiments and methods are also applicable to higher-rank cases. In training LoRA adapters, current experiments primarily focus on lower ranks (e.g., 1, 4, or 8). This choice is mainly made to ensure that the parameter storage space remains very small. To further validate the generality of our algorithm, we also tested results for r=16, which are shown in the table below.
> | Methods/Metric | BLEU | NIST | MET  | ROUGE-L | CIDEr |
> | -------------- | ---- | ---- | ---- | ------- | ----- |
> | SGD            | 65.4 | 8.07 | 40.7 | 67.0    | 2.07  |
> | Scaled GD      | 68.8 | 8.75 | 45.0 | 69.2    | 2.39  |
> | RieAdaGrad     | 70.0 | 8.81 | 46.8 | 71.9    | 2.52  |
>
> | Methods/Metric | BLEU | NIST | MET  | ROUGE-L | CIDEr |
> | -------------- | ---- | ---- | ---- | ------- | ----- |
> | AdamW          | 69.5 | 8.77 | 46.4 | 71.2    | 2.48  |
> | Scaled AdamW   | 69.8 | 8.79 | 46.5 | 71.7    | 2.51  |
> | RieAdamW       | 70.1 | 8.85 | 46.6 | 71.6    | 2.52  |
>
> The results show that our proposed RAdaGrad significantly outperforms ScaledGD on all metrics. For example, in terms of the BLEU score, our algorithm improves from 65.4 (SGD) to 70.0, and from 68.8 (ScaledGD) to 70.0. Similarly, for the ROUGE-L score, our algorithm improves from 67.0 (SGD) to 70.0, and from 69.2 (ScaledGD) to 71.9.
>
> **Comments**   **A1**  We sincerely thank you for the detailed comments on the accuracy of references and wording!
> We have carefully reviewed the issues you raised and will make comprehensive revisions in the updated manuscript to ensure the accuracy of references and the precision of our statements.
>
> **1.**  We will replace the references with more relevant ones, such as (not listed exhaustively):
>
> M. Zhang, L.-K. Tong, S. Pan, Parameter and memory efficient pretraining via low-rank Riemannian optimization, ICLR, (2025).
>
> **2.** We greatly appreciate your positive feedback on the cited book! Your suggestion to specify the exact chapters or formulas is very reasonable. We will revise the reference accordingly, citing [26, Chapter 10, 11 and 12].
>
> **3** Thank you very much for your comment on the dependency of pruning and quantization on hardware. The actual acceleration of pruning indeed depends on hardware support for sparse matrix operations. Pruned models usually result in sparse weight matrices, and sparse matrix multiplication requires hardware support for efficient sparse storage formats (e.g., CSR or CSC) and corresponding computational kernels. For example, GPUs based on NVIDIA's Ampere architecture support sparse tensor core operations, enabling performance improvements under 2:4 sparsity. However, other hardware architectures may not be able to exploit such sparsity. Similarly, Google's TPU hardware is specifically designed to support sparse matrix operations, but this requires customized compression formats and sparse acceleration kernels.
>
> Likewise, the actual acceleration and storage savings of quantization depend on hardware support and optimization for low-precision data types (e.g., INT8, BF16). Quantized models require hardware capable of efficiently performing computations with low-precision integers (e.g., INT8) or floating-point numbers (e.g., BF16). For example, NVIDIA Tensor Cores support INT8 and BF16 computations, significantly boosting the training and inference speed of quantized models. Google's TPU hardware has been optimized for BF16, enabling faster model inference without significantly reducing precision, whereas other hardware may lack similar support.
>
> **4** Regarding low-rank adaptive methods, we greatly appreciate your suggestion. We will revise our description to use the more accurate term “current mainstream low-rank adaptive methods.”
>
> We sincerely appreciate the reviewer's constructive suggestions and believe that the additional experiments, analysis, and explanations significantly improve the quality of our paper. We hope that this provides sufficient reasons to raise the score.
>
> [1] G. Becigneul and O. Ganea. Riemannian adaptive optimization methods. ICLR, 2019.
>
> [2] Geometry Aware Constrained Optimization Techniques for Deep Learning, CVPR,2018.
>
> [3] Averaging Stochastic Gradient Descent on Riemannian Manifolds, COLT 2018
>
> [4] K. Wei, J.-F. Cai, T. Chen and S. Leung. Guarantees of Riemannian optimization for low rank matrix recovery. SIMX, 2016.
>
> [5] H. Cai, J.-F. Cai, K. Wei. Accelerated alternating projections for robust principal component analysis. JMLR, 2019.
>
> [6] J.-F. Cai, J. Li, D. Xia. Provable tensor-train format tensor completion by riemannian optimization. JMLR, 2022.

---

> > ### Comment · Reviewer_oub2 · 2025-08-08
> > **Thanks for clarifications**
> >
> > Dear Authors, Thanks a lot for clarifications, I am now strongly in favor of accepting the paper. Please make sure to update and synthesize the broader literature review (and incorporate all the additional experiments/discussions).

---

> > > ### Author Response · Authors · 2025-08-09
> > >
> > > Dear reviewer,
> > >
> > > We are deeply grateful for your thoughtful feedback and strong support for our work. Your kind words and endorsement mean a great deal to us, and we sincerely appreciate the time and effort you have dedicated to reviewing our paper.
> > >
> > > We will ensure that the revised version carefully addresses your suggestions, including updating and synthesizing the broader literature review as well as incorporating all the additional experiments and discussions. Your insights have been invaluable in helping us improve the paper.
> > >
> > > Thank you again for your encouragement and guidance.
> > >
> > > Best regards,
> > > All authors of submission19911

---

### Official Review · Reviewer_S2n3 · 2025-07-07

**Clarity:** 3
**Significance:** 3
**Originality:** 3
**Rating:** 4
**Confidence:** 4

**Summary:**

The authors study memory efficient tuning of large models through low rank optimization. They propose a framework based on the optimization of the Riemannian manifold. This approach is able to eliminate the negative impact of the Jacobian condition number. Specifically, they propose RAdaGrad and RAdamW. The empirical result shows the effectiveness of the proposed method.

**Questions:**

My main question is the same as the one mentioned in the weakness section. What is the difference between the proposed method, ScaledAdam, and LoRA-RITE, which all seems like attempt to avoid the condition number problem and adopts adaptive preconditioning?

**Ethical Concerns:**

["NO or VERY MINOR ethics concerns only"]

**Final Justification:**

I appreciate the authors providing the details I ask for in the rebuttal. I believe incorporating these into the paper will make it more self-contained and make the contributions clearer to the readers. I retain my original score as it is already positive.

**Limitations:**

The limitation section feels a bit short.

**Quality:**

3

**Strengths And Weaknesses:**

Strengths:

The strength of the paper is that it seems to propose a framework that generalizes the existing methods. The empirical result also supports the effectiveness of the proposed framework.

Weakness:

However, the relation with the existing works is not discussed in detail. Specifically, the difference between the proposed method and ScaledGD and ScaledAdamW is not clearly stated, which are also methods based on Riemannian optimization. Additionally, there is LoRA-RITE[1], which also seems to avoid the Jacobian condition number, while adopts adaptive preconditioning similar to AdamW and Shampoo.

[1] Yen, J. N., Si, S., Meng, Z., Yu, F., Duvvuri, S. S., Dhillon, I. S., ... & Kumar, S. (2024). LoRA Done RITE: Robust Invariant Transformation Equilibration for LoRA Optimization. The Thirteenth International Conference on Learning Representations (ICLR 2025)

---

> ### Author Rebuttal · Authors · 2025-07-27
>
> **W1/Q1.**  However, the relation with the existing works is not discussed in detail. Specifically, the difference between the proposed method and ScaledGD and ScaledAdamW is not clearly stated, which are also methods based on Riemannian optimization. Additionally, there is LoRA-RITE[1], which also seems to avoid the Jacobian condition number, while adopts adaptive preconditioning similar to AdamW and Shampoo.
>
> **A1.** Thank you for pointing out that our discussion of existing work is not sufficiently thorough. We greatly value your feedback and will include a more detailed comparison and discussion in the revised manuscript. Below, we provide a specific explanation of the connections and differences between our algorithm and related works:
>
> **[1] Differences from ScaledGD and ScaledAdamW.**
>
> Although ScaledGD and ScaledAdamW are also based on preconditioning methods for Riemannian optimization, there are several key differences between their algorithms and ours:
>
> **1.**  **Optimization framework for weight updates.** According to [1, Section 3], the weight matrix update in ScaledGD can be reformulated as
>
> $$
> \begin{aligned}
> X_{t+1} &\approx  X_t - \eta (\nabla_{X_t} L) A_t^{T} (A_tA_t^{T})^{-1}A_t - \eta B_t(B_t^TB_t)^{-1}B_t^T(\nabla_{X_t}L) =X_t - \eta Proj_{row(A_t)} (\nabla_{X_t} L) - \eta Proj_{col(B_t)}(\nabla_{X_t} L)^T.
> \end{aligned}
> $$
>
> The update in our algorithm for the whole weight matrix on the tangent space $\mathbb{T}_t$ can be expressed as
>
> $$
> W_t - \gamma_t \mathcal{P}_{\mathbb{T}_t}^{(g)}(Z_t) = W_t - \tilde{U}_t\tilde{U}_t^{T}L_t^{-1/4}Z - Z R_t^{-1/4}\tilde{V}_t \tilde{V}_t^{T} + \tilde{U}_t\tilde{T}_t^{T}L_t^{-1/4}ZR_t^{-1/4}\tilde{V}_t \tilde{V}_t^{T}
> $$
>
> $$
> \qquad \qquad \qquad \quad =W_t - \tilde{U}_t\tilde{U}_t^{T}G_tR_t^{-1/4} - L_t^{-1/4} G_t \tilde{V}_t \tilde{V}_t^{T} + \tilde{U}_t\tilde{T}_t^{T} G_t \tilde{V}_t \tilde{V}_t^{T}
> $$
>
> $$
> \qquad \qquad \qquad \quad =W_t  - Proj_{row} (G_t) - Proj_{col}(G_t) + Proj_{row}Proj_{col}(G_t),
> $$
> where $Z_t = L_t^{-1/4}G_tR_t^{-1/4}$. From the update rules of the two algorithms, it is evident that ScaledGD applies projections of the gradient onto both the row and column spaces twice for overlapping gradient components $Proj_{row}Proj_{col}(G_t)$. In contrast, our algorithm applies these projections only once. As a result, ScaledGD contains redundant projected gradient information, which may lead to excessive updates, deviations in the optimization path, and ultimately inferior training performance for the network. This also adversely affects the generative results of language models, as demonstrated by our experimental results.
>
> **2. Different methods for mitigating the Jacobian condition number.**  ScaledGD, based on a decomposition formulation, essentially solves the optimization problem on the quotient manifold M/ ∼, where each element [x]={y∈M:y∼x} represents an equivalence class. ScaledGD defines a new metric on the quotient manifold [x]=(A,B) as follows
>
> $$
> g [x] (\eta [x],\xi [x] ) = \langle \eta_A, \xi_A B^TB\rangle + \langle \eta B, \xi_{B} A^TA \rangle.
> $$
>
> Under this metric, the update formulas for $A$ and $B$ can be derived as
>
> $$
> (\frac{\partial}{\partial A}) \to (\frac{\partial}{\partial A}) (B^TB)^{-1}, \quad (\frac{\partial}{\partial B}) \to (\frac{\partial}{\partial B}) (A^TA)^{-1}.
> $$
>
> From these formulas, we observe that the new gradients in ScaledGD for the low-rank factors $A$ and $B$ are preconditioned by $(B^TB)^{-1}$ and $(A^TA)^{-1}$, respectively, which partially alleviate the effects of the Jacobi condition number. However, as noted earlier, ScaledGD does not fully eliminate the influence of the $\mathcal{J}_{\mathcal{G}}^*$ condition number. More importantly, it does not address the condition number of the Hessian $\nabla^2 \ell(W)$.
>
> As you pointed out, our main contribution lies in proposing a unified framework for Riemannian gradient descent algorithms for the entire weight matrix update. By selecting appropriate metrics and retraction operators, we achieve a more efficient algorithm. Specifically, our approach targets the $W$ update, thereby completely avoiding the influence of the $\mathcal{J}_{\mathcal{G}}^*$ condition number. Furthermore, we introduce the following metric, which combines AdaGrad and Shampoo
>
> $$
> \langle X, Y \rangle_{g_t} = \langle X, A_t Y \rangle = \langle X, L_t^{1/4} Y R_t^{1/4} \rangle,
> $$
>
> where $L_t = \epsilon I + \sum_{t} diag(G_t G_t^T)$ and $R_t = \epsilon I + \sum_{t} diag(G_t^T G_t)$.  Under this new metric, we obtain a newly preconditioned Riemannian gradient $\tilde{\mathcal{P}}_{\mathbb{T}}(L_t^{-1/4}G_t R_t^{-1/4})$, which exhibits isotropic properties, effectively mitigating the influence of the Hessian condition number. In summary, the metric we select is more effective for updating low-rank matrix weights. All of our experimental results strongly support this conclusion.
>
> **3. Different convergence rates.** Further referencing the convergence analysis in [1], the convergence of ScaledGD relies on the following assumptions
>
> $$
> (1−δ_r)∥M∥_F^2 ≤∥AM∥_F^2 ≤(1+δ_r)∥M∥_F^2.
> $$
> Based on these assumptions, the convergence rate of ScaledGD depends on $\kappa= 2 \frac{1+δ_r}{1−δ_r}$. When $δ_r$ is very small,  $\kappa \approx 2$. We analyze the convergence of our algorithm under the same assumptions. Preliminary calculations show that the convergence rate of our algorithm depends on the condition number $\kappa= \frac{1+δ_r}{1−δ_r}$, when $δ_r$  is very small, $\kappa \approx 1$. Clearly, under the same assumptions, our algorithm demonstrates better convergence than ScaledGD. For ScaledAdamW, similar conclusions apply as with ScaledGD, since the primary difference lies in the optimizer used.
>
> [1] F. Zhang and M. Pilanci. Riemannian preconditioned LORA for fine-tuning foundation models. In Proceedings of the 41st International Conference on Machine Learning (ICML), 2024.
>
> **[2] Differences from LoRA-RITE.**    We will cite LoRA-RITE [1] in the revised version of our paper. Below, we elaborate on the key differences between our algorithm and LoRA-RITE, along with their potential implications, while highlighting the advantages of our approach.
>
> Although LoRA-RITE achieves transformation invariance by introducing preconditioners, its core framework remains based on factorization for updating low-rank factors. Specifically, LoRA-RITE designs adaptive preconditioners and incorporates first- and second-order momentum using gradients of the low-rank factors. However, its gradient updates rely on the pseudo-inverse of the low-rank factor gradients to approximate the inverse of the Hessian. Due to the over-parameterized nature of low-rank factorization, these gradient matrices are often not invertible. As a result, the use of pseudo-inverses leads to the loss of gradient information in directions orthogonal to the pseudo-inverse, introducing errors that can affect the optimization path and final performance.
>
> This highlights the fundamental difference between our algorithm and factorization-based methods. Our method directly utilizes the gradients of the entire weight matrix $W$ to construct the preconditioner, first-order momentum, and second-order momentum. Since we do not rely on low-rank factorization, our gradient matrices are invertible, allowing us to directly and more effectively approximate the inverse of the Hessian matrix. This avoids the information loss and potential errors introduced by pseudo-inverses in LoRA-RITE. By leveraging the full weight matrix gradients, our approach demonstrates potential advantages in optimization efficiency and final performance.
>
> We sincerely appreciate the reviewer's constructive suggestions and believe that the additional experiments, analysis, and explanations significantly improve the quality of our paper. We hope that this provides sufficient reasons to raise the score.
>
> [1] Yen, J. N., Si, S., Meng, Z., Yu, F., Duvvuri, S. S., Dhillon, I. S., ... & Kumar, S. LoRA Done RITE: Robust Invariant Transformation Equilibration for LoRA Optimization. The Thirteenth International Conference on Learning Representations (ICLR 2025)

---

> > ### Comment · Reviewer_S2n3 · 2025-08-08
> >
> > I appreciate the authors providing the details I ask for in the rebuttal.
> > I believe incorporating these into the paper will make it more self-contained and make the contributions clearer to the readers.
> > I retain my original score as it is already positive.

---

> > > ### Author Response · Authors · 2025-08-09
> > >
> > > We sincerely appreciate your valuable feedback during the review process. We are thrilled to have addressed your concerns and further enhanced the quality and clarity of the paper.
> > >
> > > We also greatly appreciate you maintaining the positive score for our paper, which serves as an important encouragement for our research efforts. We will carefully incorporate these improvements into the final version to ensure the paper is more self-contained and easier for readers to understand.
> > >
> > > Thank you once again for your support and acknowledgment of our work!

---

### Official Review · Reviewer_eKS4 · 2025-07-09

**Clarity:** 3
**Significance:** 2
**Originality:** 2
**Rating:** 2
**Confidence:** 4

**Summary:**

This paper proposes a Riemannian optimization algorithm for the problem of low rank training of deep networks. The authors claim that existing methods are vulnerable to the condition number of the jacobian operator for the factor and their proposed method avoids this trap. The authors derive riemannian equivalents of adamW and adagrad by proposing alternative riemannian metrics. The paper empirically demonstrates improved performance in LM finetuning and pre-training tasks.

**Questions:**

See weaknesses

**Ethical Concerns:**

["NO or VERY MINOR ethics concerns only"]

**Quality:**

2

**Strengths And Weaknesses:**

Strengths: The paper studies an important problem of LoRA finetuning. The empirical results show that their training algorithms improve performance.

Weaknesses: This paper fails to adequately demonstrate the existence of a convergence problem in LoRA finetuning, and further demonstrate that their method adequately addresses the problem.

1. The authors do not include any examples of failed convergence in training low rank factorization models, and only argue that the condition number of W is a problem through a derivation of the hessian. They do not compare convergence rates that the condition number of the jacobian plays a factor in the poor convergence of SGD in the factored parameterization.

2. The proposed method is Riemannian gradient descent/Adam using the manifold of low rank matrices. This does not improve on the time complexity of the factorized parameterization, and might involve more computation since an SVD is required at each step.

3. The empirical evaluation only establishes improved performance in terms of BLEU, etc. The authors do not show that their convergence times are better.

4. Prior work has already developed riemannian adaptive gradient methods, which the authors do not cite. https://arxiv.org/abs/1810.00760

---

> ### Author Rebuttal · Authors · 2025-07-26
>
> **W1/Q1.**  **A1.**  Thank you for your valuable feedback. We greatly appreciate your comments and believe they will help improve our work.
>
> **[1]**  Although the current version does not include a complete theoretical analysis due to time constraints, this omission is not due to any fundamental difficulty in the derivations. In fact, the theoretical analysis framework for our algorithm has already been preliminarily established, and its derivation process is more straightforward compared to factorization-based algorithms. This is because our algorithm updates the weight matrix $W$ as a whole, involving only $\nabla \ell (W)$, without needing to handle the complexities introduced by the factorization form, such as $\mathcal{J}_{\mathcal{G}}^*(P, Q)$. Thus, by applying standard smoothness assumptions, such as Lipschitz continuity, to $\nabla \ell (W)$, it is relatively easy to establish the convergence analysis and derive the convergence rate of the algorithm.
>
> **[2]** In the supplementary material, we presented experiments on fine-tuning a mix-of-show language model. Figures 3 and 4 illustrate the final results and intermediate results from different training epochs, respectively. Figure 4 shows the model's generated outputs at  500th and 2000th. The results clearly demonstrate that at 500th epoch, scaledGD and scaledAdamW are unable to generate any meaningful information, whereas our algorithm, RAdaGrad, has already generated images highly relevant to the prompt ``A woman with long hair in Simpsons style". By 2000th epoch, our algorithm has produced images that matched the prompt very well, while the images generated by scaledGD and scaledAdamW still contained significant flaws and were far from usable.
>
> These experimental results clearly indicate that our algorithm can generate usable images with far fewer training epochs. In our responses to W2/Q2, we also provided the loss. The table demonstrates that while the training time for our algorithm is comparable to that of other algorithms, the final image quality produced by our algorithm is the highest, with the best alignment to the input prompts, which is critical for generative networks. Given the same training time, the superior image quality generated by our algorithm is strong evidence of its higher efficiency.
>
> **[3]** During the experiments, we observe that using standard decomposition methods can lead to pathological optimization when small singular values are present. This is a characteristic of the low-rank matrix manifold itself, where the curvature of the Hessian matrix is inversely proportional to the smallest singular value of the weight matrix. To evaluate the efficiency of our algorithm under such conditions, we chose a larger  r=16 to ensure that the Hessian matrix has a high condition number. The results show that our proposed RAdaGrad significantly outperforms ScaledGD on all metrics. For example, in terms of the BLEU score, our algorithm improves from 65.4 (SGD) to 70.0, and from 68.8 (ScaledGD) to 70.0. Similarly, for the ROUGE-L score, our algorithm improves from 67.0 (SGD) to 70.0, and from 69.2 (ScaledGD) to 71.9.
>
> | Methods/Metric | BLEU | NIST | MET  | ROUGE-L | CIDEr |
> | -------------- | ---- | ---- | ---- | ------- | ----- |
> | SGD            | 65.4 | 8.07 | 40.7 | 67.0    | 2.07  |
> | Scaled GD      | 68.8 | 8.75 | 45.0 | 69.2    | 2.39  |
> | RieAdaGrad     | 70.0 | 8.81 | 46.8 | 71.9    | 2.52  |
>
> **W2/Q2.**  **A2.**  Thank you for your insightful question. We greatly value your feedback and would like to clarify the computational aspects of our proposed method.
>
> **[1]** Compared to factorization-based algorithms, our method does indeed perform an SVD computation at each step. However, as analyzed in Sections 2 and 3 of the supplementary materials, the computational complexity of performing SVD on $\mathcal{H}_r(Z_t)$ at each step is $\mathcal{O}(r^3)$, depending only on the rank $r$. LoRA fine-tuning typically selects a very small rank $r$, which makes the computational cost of SVD extremely low. In practice, this computational cost is negligible and has an almost imperceptible impact on the overall efficiency.
>
> **[2]** To further validate the efficiency of our algorithm, we compare the training loss of all algorithms under the same number of iterations (i.e., the same number of epochs), as shown in the table below. The experimental results demonstrate that our algorithm achieves the fastest loss reduction during training. For example, the loss trend table shows that at the 2000th iteration, our algorithm reduces the loss to 2.73, while the losses of other algorithms remain above 2.80. By the 18000th iteration, our algorithm further reduces the loss to 2.59, whereas the losses of other algorithms remain above 2.60. This indicates that, given the same training time, our algorithm achieves significantly lower loss values, resulting in superior overall performance. Furthermore, as demonstrated in Figures 3 and 4 of the supplementary materials, the output quality generated by our algorithm is also superior. These results highlight the effectiveness of our method, especially for network training based on limited data, where efficient and effective learning is crucial.
>
> Training Average Loss:
>
> | Iteration  | 2000 | 4000 | 6000 | 8000 | 10000 | 12000 | 14000 | 16000 | 18000 | 20000 | 22000 |
> | -----------------  | ---- | ---- | ---- | ---- | ----- | ----- | ----- | ----- | ----- | ----- | ----- |
> | SGD                 | 2.89 | 2.89 | 2.81 | 2.86 | 2.83  | 2.81  | 2.82  | 2.80  | 2.76  | 2.75  | 2.69  |
> | scaled GD           | 2.80 | 2.79 | 2.71 | 2.75 | 2.72  | 2.70  | 2.72  | 2.70  | 2.67  | 2.66  | 2.60  |
> | plain RGD          | 2.80 | 2.78 | 2.69 | 2.73 | 2.70  | 2.67  | 2.70  | 2.67  | 2.64  | 2.64  | 2.59  |
> | RAdaGrad (ours)     | 2.73 | 2.72 | 2.63 | 2.68 | 2.65  | 2.63  | 2.64  | 2.62  | 2.59  | 2.59  | 2.53  |
>
> **W3/Q3.**   **A3.**  Thank you for highlighting this important evaluation metric, which will improve the presentation of our work. Our experimental evaluation focuses on improved performance metrics, which directly demonstrate better results within the same number of iterations and thus, greater algorithm efficiency. To further validate the effectiveness, we analyze the loss reduction trends (see the table in W2/Q2). The results show that our method achieves faster loss reduction compared to other algorithms. For example, at the 2000th iteration, the loss value of our method is lower than that of the baselines. This demonstrates that our algorithm requires fewer iterations to achieve good results, which is particularly crucial for low-data or few-shot learning. Additionally, Figures 3 and 4 in the supplementary materials further support that, within the same training time, our method can generate outputs that align better with the input prompts. All these experimental results clearly show that our algorithm converges more quickly and effectively.
>
> **W4/Q4.**  **A4.**  Thank you very much for bringing this important reference to our attention. This paper is closely related to our research, and we will revise our manuscript to cite and discuss [1]. In this work, the authors extend adaptive optimization methods to Riemannian manifolds, particularly for product manifolds. While both our algorithm and [1] introduce adaptive methods, there are several key differences
>
> **[1]**  [1] generalizes adaptive methods to Riemannian manifolds, whereas our algorithm specifically introduces an adaptive method tailored for low-rank matrix manifolds. This allows our algorithm to exploit the compact and smooth structure of low-rank matrix manifolds, further enhancing its efficiency. For example, our algorithm consists of three steps
>
> **1.**  $M_t = W_t - \gamma_t L_t^{-1/4} G_t R_t^{-1/4}.$
>
> **2.** $Z_t= \mathcal{P}_{\mathbb{T}_t}^{g} (W_t - \gamma_t L_t^{-1/4} G_t R_t^{-1/4}). $
>
> **3.** $W_{t+1} =\mathcal{H}_r (Z_t) =\mathcal{H}_r(W_t -  \gamma_t\mathcal{P} (L_t^{-1/4} G_t R_t^{-1/4})).$
>
> We parameterize the tangent space of the low-rank matrix manifold at a point, then steps 2 and 3 can be combined for computation (see the details in Sections 2 and 3 of the supplementary materials). This allows us to perform SVD on only a $2r\times 2r$ matrix, reducing the computational complexity of SVD to $\mathcal{O}(r^3)$. In contrast, the algorithm in [1] does not include a tangent space projection (Step 2), resulting in an SVD computational complexity of $\mathcal{O}(rn^2)$. When $r$ is small, the computational efficiency of our algorithm is significantly higher than that of the algorithm in [1].
>
> **[2]** More importantly, we propose a unified Riemannian gradient framework $W_{t+1} = \mathcal{R}(W_t - \gamma_t \nabla_{\mathcal{M}_r} \ell(W_t)$, which provides new insights for designing efficient algorithms. As shown in 3.1, when the standard metric is chosen, our method reduces to the plain RGD [2]. In 3.2, we select a metric that combines AdaGrad and Shampoo, ensuring isotropy of gradients and mitigating the effects of the Hessian condition number, resulting in efficient RAdaGrad. This demonstrates that selecting more effective retraction and metrics can lead to more efficient algorithms. This idea is critical for designing efficient algorithms and is particularly inspiring for developing efficient fine-tuning methods for large language models. We hope this heuristic idea can stimulate the design of even more efficient algorithms.
>
> We sincerely appreciate the reviewer's constructive suggestions and believe that the additional experiments, analysis, and explanations significantly improve the quality of our paper. We hope that this provides sufficient reasons to raise the score.
>
> [1] G. Becigneul and O. Ganea. Riemannian adaptive optimization methods. ICLR, 2019.
>
> [2] K. Wei, J.-F. Cai, T. Chen and S. Leung. Guarantees of Riemannian optimization for low rank matrix recovery.
> SIAM J. Matrix Anal. Appl., 37(3):1198–1222, 2016.

---

### Note · Authors · 2025-08-15

Dear Program Chairs, Senior Area Chairs, and Area Chairs,

We sincerely thank you for giving us the opportunity to submit this "Final Remark". We would like to take this opportunity to once again emphasize the key contributions of our paper:

Our paper identifies that factorization-based low-rank matrix algorithms suffer from non-uniqueness and redundancy, whose convergence rates are constrained by the condition numbers of both the Hessian matrix $\nabla^2 \ell(W)$ and the Jacobian matrix $J_{G}(P, Q)$. To address this limitation, we propose a novel Riemannian Gradient Descent framework, which directly optimizes the low-rank matrix $W$ on a fixed-rank manifold, eliminating the redundancy caused by factorization. Consequently, the convergence rate of RGD depends solely on $\nabla^2 \ell(W)$ and is independent of the condition number of $W$. While RGD is comparable to factorization-based methods in computational complexity and memory usage, it offers notable advantages. Furthermore, leveraging the flexibility of Riemannian metrics, we define different metrics on the manifold, enabling the extension of classical adaptive learning rate and momentum algorithms to Riemannian manifolds. Within this framework, we introduce two new algorithms: RAdaGrad and RAdamW. These algorithms preserve the adaptivity of AdaGrad and AdamW while better aligning with the structure of low-rank matrix optimization. Extensive experiments demonstrate that both RAdamW and RAdaGrad consistently outperform existing SOTA methods.

The reviewers recognized our contributions and provided valuable feedback, along with some concerns that we addressed in detail to further improve our paper. Four out of five reviewers expressed support or strong support:

*$\cdot$* Reviewer VP56 supports the paper.

*$\cdot$* Reviewer 9j4S actively engages in discussions and indicates a willingness to raise the score.

*$\cdot$* Reviewer S2n3 maintain a positive score.

*$\cdot$* Reviewer oub2 strongly supports the paper after reviewing our rebuttal.

Reviewer eks4's comments overlap with other four reviewers. We have addressed these concerns in detail and received their support. As such, we are confident that the comments from Reviewer eks4 have also been effectively addressed.

We are deeply grateful to all the reviewers for their valuable and thoughtful feedback, which has helped us refine and improve our work.

Thank you for your time and consideration.

Sincerely,

Authors of submission 19911

---

### Decision · Program_Chairs · 2025-09-17

**Decision:**

Accept (poster)

**Comment:**

Following an active discussion between authors and reviewers, all reviewers recommended acceptance except for one, who did not reply to the authors or participated in the discussion. I think the authors provided a satisfactory reply to this reviewer. In summary, the paper was deemed an interesting contribution to the timely area of estimating or adapting neural nets with low-rank matrices.

A further comment from the AC: in line 33 and 40, an important problem in low-rank DNN compression is to learn the rank at each layer. A relevant citation: "Low-rank compression of neural nets: Learning the rank of each layer", CVPR 2020. This work uses gradients on the full matrix but penalizes it towards a low-rank matrix which is the result of a SVD at each step, but whose factors are not updated. How does the discussion in lines 39-65 apply here? Although the authors' method is different, there are similarities in the goals to directly update the entire low-rank weight matrix rather than its low-rank factors, and to avoid the non-uniqueness and redundancy introduced by factorization.